# Molecular correlates of muscle spindle and Golgi tendon organ afferents

Katherine M. Oliver[1,2,6], Danny M. Florez-Paz [2,3,6], Tudor Constantin Badea [4,5], George Z. Mentis[1,2,3], Vilas Menon [1] & Joriene C. de Nooij [1,2✉]

Proprioceptive feedback mainly derives from groups Ia and II muscle spindle (MS) afferents and group Ib Golgi tendon organ (GTO) afferents, but the molecular correlates of these three afferent subtypes remain unknown. We performed single cell RNA sequencing of genetically identified adult proprioceptors and uncovered five molecularly distinct neuronal clusters. Validation of cluster-specific transcripts in dorsal root ganglia and skeletal muscle demonstrates that two of these clusters correspond to group Ia MS afferents and group Ib GTO afferent proprioceptors, respectively, and suggest that the remaining clusters could represent group II MS afferents. Lineage analysis between proprioceptor transcriptomes at different developmental stages provides evidence that proprioceptor subtype identities emerge late in development. Together, our data provide comprehensive molecular signatures for groups Ia and II MS afferents and group Ib GTO afferents, enabling genetic interrogation of the role of individual proprioceptor subtypes in regulating motor output.

[1] Department of Neurology, Vagelos College of Physicians and Surgeons, Columbia University Medical Center, New York, NY, USA. [2] Columbia University Motor Neuron Center, Columbia University Medical Center, New York, NY, USA. [3] Department of Pathology, Columbia University Medical Center, New York, NY, USA. [4] Retinal Circuit Development and Genetics Unit, National Eye Institute, Bethesda, MD, USA. [5] Research and Development Institute, Transylvania University of Brasov, Faculty of Medicine, Brasov, Romania. [6]These authors contributed equally: Katherine M. Oliver, Danny M. Florez-Paz. ✉email: sd382@cumc.columbia.edu

Sensory feedback from muscle, skin, and joints is critical for the normal execution of voluntary motor tasks. Collectively referred to as the proprioceptive sense, this afferent information informs the central nervous system (CNS) on the position of the body and limbs in space[1–3]. Many decades of study of the intramuscular sense organs and their afferent innervation have led to predictions on how proprioceptive feedback integrates with other sensory modalities to influence central motor circuits and calibrate motor output, yet most of these inferences await further corroboration in behaving animals[4–6]. At present, such studies remain challenging due to the lack of genetic access to individual proprioceptor subtypes.

Proprioceptive feedback derives in large part from specialized mechanoreceptive organs in skeletal muscle. Extensive anatomical and physiological analysis have revealed two types of muscle receptors: muscle spindles and Golgi tendon organs[7]. Muscle spindle (MS) mechanoreceptors are considered the main drivers for the sense of limb position and movement (kinesthetic sense)[8,9]. They are embedded within the belly of skeletal muscles and consist of encapsulated intrafusal muscle fibers that are typically innervated by one primary (group Ia) and several secondary (group II) proprioceptive sensory neurons (pSNs) (Fig. 1a)[7,10].

Both types of afferents are responsive to stretch of the intrafusal fibers, such that voluntary or passive changes in limb position (i.e. muscle length) result in increased or decreased firing rates[11]. Unique among sensory organs, MSs are subject to (CNS-directed) efferent motor control through dynamic and static gamma motor neurons (γMN) which innervate the contractile polar ends of the intrafusal muscle fibers and effectively set the gain for group Ia/II afferent discharge frequency[12,13]. Despite supplying the same sensory end organ, group Ia and group II afferents exhibit distinct intra-spindle innervation patterns, activation thresholds, and conduction velocities, features that appear to render them biased to qualitatively different information of muscle stretch. For instance, while both group Ia and II afferents relay changes in static muscle length, the dynamic sensitivity of group Ia afferents enables them to also signal the rate of change in muscle length[14,15]. Presently, to what extent these properties are intrinsic to the neurons or to their association with different intrafusal muscle fibers remains poorly understood.

In contrast to MSs, Golgi tendon organs (GTOs) are mechanoreceptive organs that are concerned with the sense of effort and respond to changes in muscle load[16]. GTO receptors are located at the myotendinous junction of skeletal muscles and

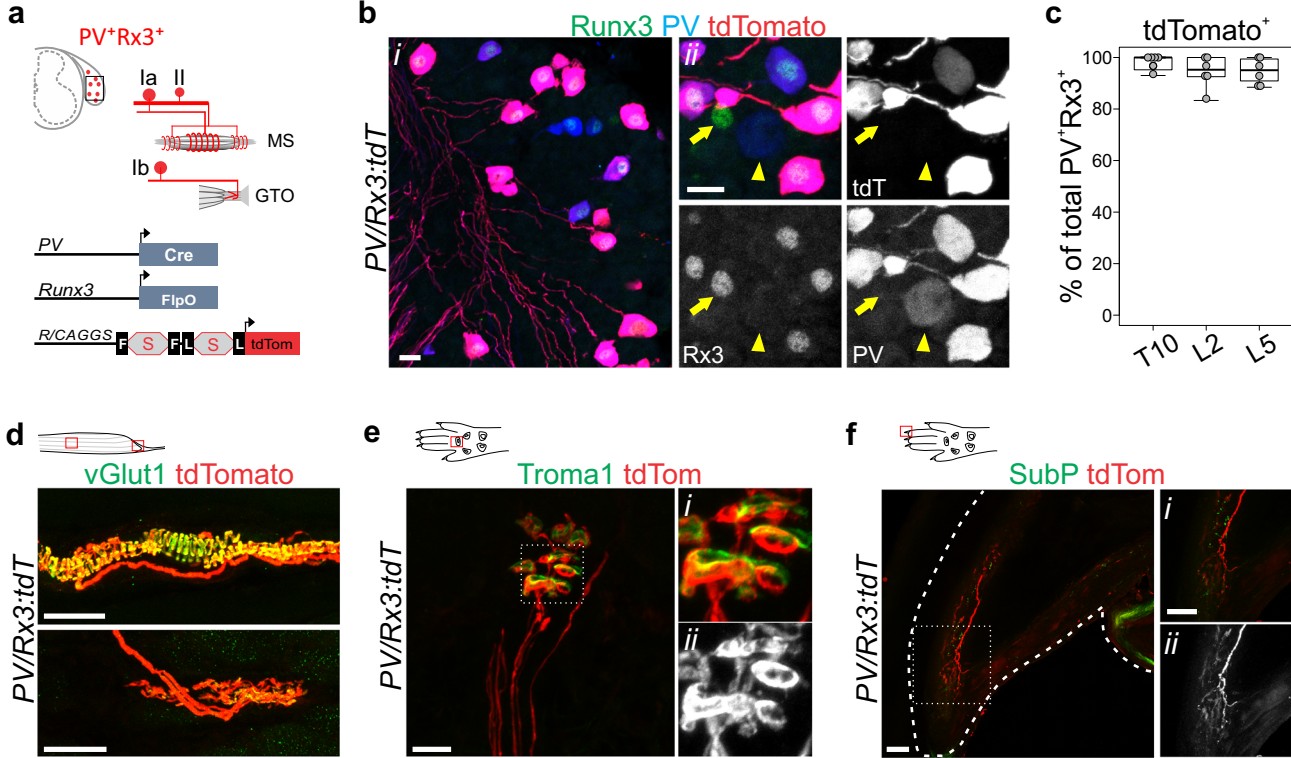

**Fig. 1 An intersectional genetic labeling strategy for proprioceptive muscle afferents. a** Schematic rendering of the groups Ia, II and Ib proprioceptor subtypes labeled in mice carrying a *PV:Cre*, *Rx3:FlpO*, and the double stop *Ai65:tdTomato* allele (*PVRx3:tdT*). In sensory neurons that co-express PV and Rx3, the dual activity of Cre and FlpO recombinases will allow expression of the tdTomato reporter. **b** Expression of Runx3, PV, and tdTomato in DRG at p2 indicates efficient induction of the tdTomato reporter in PV⁺Rx3⁺ sensory neurons, but not in PV⁺Rx3ᵒᶠᶠ (arrowhead in ii) or PVᵒᶠᶠRx3⁺ (arrow in ii) neurons (i and ii are independent images). **c** Efficiency of tdTomato reporter induction in T10, L2, and L5 DRG in p2 *PVRx3:tdT* mice. Mean percentage ± S.E.M. of tdT⁺PV⁺Rx3⁺ neurons (of all PV⁺Rx3⁺) per ganglia for T10: 98.0 ± 1.3%, for L2: 94.4 ± 2.5%, for L5: 94.8 ± 1.9%; *n* = 6 DRG per segmental level. Number of sections per T10 DRG: 7, 6, 6, 7, 7, 8; Number of sections per L2 DRG: 7, 3, 6, 5, 8, 7; Number of sections per L5 DRG: 9, 8, 10, 6, 7, 10. **d** tdTomato expression in vGlut1⁺ muscle spindle (MS) and Golgi tendon organ (GTO) afferent terminals in p10 EDL muscle of *PVRx3:tdT* animals. Diagram depicts muscle with red boxes showing areas in images below. Expression of tdTomato in Merkel cell afferents present in hind paw glabrous skin (**e**) and in SubstanceP(SubP)ᵒᶠᶠ sensory neurons that appear to innervate the nail of forepaw digits (**f**) of adult (≥p56) *PVRx3:tdT* mice. Diagram depicts foot paw with red boxes showing areas in images below. In **e**, Merkel cells are indicated by expression of Troma1. i, ii show higher magnifications of boxed areas in **e**, **f**; ii shows tdTomato expression. Similar data obtained from at least three (**b**, **d**, **e**) biological replicates. In boxplot (**c**), boxes indicate medians and 25th/75th percentiles, and whiskers extend to the furthest point <1.5 standard deviations from the mean. Scale 20 (**b**) or 50 (**d–f**) µm.

are innervated by a single group Ib afferent. Group Ib afferents have similar activation thresholds and conduction velocities as group Ia MS afferents. They show little activation to muscle stretch but are highly sensitive to contraction of the motor units that are in series with the tendon organ they innervate (Fig. 1a)[17]. Group Ib afferents also exhibit dynamic sensitivity and can signal rapid changes in contractile force[18]. Due to their physiological similarities to Ia MS afferents, selective activation of group Ib GTO afferents has remained challenging, leaving a number of uncertainties regarding their central pathways and physiological role in motor control.

Underscoring their functional differences, MS and GTO afferents engage with different local reflex circuits and projection neurons[15,19,20]. Most notably, MS afferents (and in particular group Ia afferents), but not GTO afferents, establish mono-synaptic contacts onto homonymous αMNs that innervate the same muscle target, while GTO afferents mostly inhibit homon-ymous MNs through di/tri-synaptic connections[21,22]. Most other spinal excitatory or inhibitory reflex circuits receive overlapping input from groups Ia, II, and Ib afferents yet are biased towards a given proprioceptor subtype or combination of these subtypes[19,23,24]. Similarly, information from all three classes of afferents is relayed to higher brain circuits through projection neurons of the DSCT and VSCT but the influence of group Ia, II or Ib feedback appears to be distributed differentially across the different tracts[20]. Thus, the three proprioceptive muscle afferents subtypes (groups Ia, II, and Ib) innervate one of two anatomically and functionally distinct mechanoreceptive organs, exhibit diverse physiological properties and engage with different central targets. Yet, over a century and a half after these afferents were first described (as referenced in ref. [7]), a definitive molecular signature for functionally distinct proprioceptive muscle afferents remains to be established.

Advances in next generation RNA sequencing (RNAseq) technologies have vastly expanded our general insight into the molecular diversity of adult somatosensory neuron classes, including proprioceptors[25,26]. Nevertheless, since proprioceptors make up only ~10% of the total DRG population, under-sampling of this neuronal class has thus far precluded a molecular classi-fication of MS or GTO afferent subtypes. We here devised an intersectional genetic strategy to label and isolate muscle pro-prioceptors, which enabled us to maximize their sampling density for single cell transcriptome analysis. Single cell RNAseq of adult proprioceptors and subsequent bioinformatics analysis identified five molecularly distinct neuronal clusters. Validation of these neuronal clusters indicates that one of these clusters corresponds to afferents that morphologically resemble group Ia MS afferent. pSN neurons that belong to a second cluster invariably associate with GTO sensory endings, indicating they correspond to group Ib afferents. By inference, we postulate that the remaining iden-tified molecular clusters may correspond to group II MS afferents. Additional single cell RNAseq analysis of proprioceptors at earlier developmental stages enabled a lineage analysis between tran-scripts that identify MS or GTO proprioceptor subtypes in the adult with these transcripts at earlier developmental time points. These developmental analyses support the idea that proprioceptor subtype identity is established at a late developmental stage, after the afferents innervate their nascent receptor targets in muscle. Thus, these data offer valuable insights into the functional diversity of proprioceptive MS and GTO afferents, the molecular underpinnings of their physiological properties, and the mole-cular pathways through which distinct proprioceptor types emerge. Moreover, the identification of molecules that unam-biguously delineate MS and GTO afferent subtypes offers a means to genetically dissect the relative contributions of these afferents in the execution of coordinated motor behavior.

## Results

**Intersectional genetic labeling of proprioceptive muscle affer-ents.** Proprioceptive sensory neurons in DRG can be identified by the co-expression of Parvalbumin (PV) and the Runt-domain transcription factor Runx3 (Rx3)[27,28]. In contrast, the singular expression of these molecules marks subsets of low threshold cutaneous mechanoreceptive afferents with rapidly or slowly adapting response properties, respectively[27,29]. To permit the selective isolation of $PV^+Rx3^+$ proprioceptors from these other types of mechanoreceptors, we generated a Rx3:FlpO allele, thus enabling an intersectional genetic strategy in conjunction with a previously generated PV:Cre allele (Fig. 1a and Supplementary Fig. 1a)[30]. In assessing the accuracy and efficacy of Rx3:FlpO-mediated reporter expression in the context of a FlpO-dependent GFP reporter mice (RCE:FRT)[31], we find that, on average, $91.2 \pm 1.9\%$ of $Rx3^+$ DRG neurons express GFP following FlpO-mediated excision of the transcriptional stop ($n = 6$ L2 DRG) (Supplementary Fig. 1c–g). Induction of GFP expression appears slightly more efficient in $Rx3^+PV^+$ pSNs compared to $Rx3^+PV^{off}$ neurons (presumed Merkel cell afferents) (Supplementary Fig. 1h, i)[27]. Consistent with the notion that Rx3 is initially more widely expressed in embryonic (e) sensory progenitors (e11–e13.5) (Supplementary Fig. 1b), at postnatal day (p) 0 the number of $GFP^+$ neurons exceeded the number of $Rx3^+$ neurons ($30.9 \pm 2.1\%$ of $Islet1^+GFP^+$ neurons lack expression of Rx3; $n = 6$ DRG) (Supplementary Fig. 1d, g). We find, however, that these $Rx3^{off}GFP^+$ neurons (in which GFP expression is activated by the transient developmental expression of Rx3) do not express PV, suggesting that this population does not include $PV^+$ cutaneous mechanoreceptors (Supplementary Fig. 1e). In addition, GFP reporter expression is not detected in spinal interneurons or motor neurons (Supplementary Fig. 1j). Likewise, aside from the expected proprioceptor sensory endings, within muscle only a few $GFP^+$ satellite cells were detected (Supplementary Fig. 1k). These data indicate that, at least within DRG, spinal cord, and muscle, the Rx3:FlpO reporter faithfully reflects endogenous Rx3 expres-sion and presents a useful tool that enables genetic access to $Rx3^+$ DRG sensory neurons.

We next examined the Rx3:FlpO allele in the context of a PV:Cre driver and the Ai65:double stop: tdTomato$^+$ reporter (here-after PVRx3:tdT)[30,32]. In PVRx3:tdT mice, expression of tdTomato is expected to be limited to DRG neurons that express both PV and Rx3 at any one stage of their development (Fig. 1a and Supplementary Fig. 2a–e). Indeed, we find that tdT invariably coincides with $PV^+Rx3^+$ neurons at all segmental levels analyzed (T10, L2, L5) and is not observed in neurons that express only Rx3 or PV (Fig. 1b). At p2, an average of $95.7 \pm 1.3\%$ of $PV^+RX3^+$ neurons co-express tdT ($n = 18$ DRG) (Fig. 1b, c). Consistent with these observations, we find that tdT$^+$ afferents associate with MS and GTO sensory end organs in muscle (Fig. 1d). Expression of tdT is, however, also observed in a few Rx3$^+$ neurons with very low levels of PV expression ($PV^{low}Rx3^+$) (Supplementary Fig. 2c–f), possibly representing a different neuronal subset that either transiently expresses PV, or that express PV at levels much lower than typically observed in proprioceptive muscle afferents. Indeed, while expression of tdTomato in spinal cord is strictly limited to sensory afferent projections (i.e. no expression is observed in spinal neurons or microglia), we noted a few afferents that entered the spinal cord laterally (as opposed to the medial spinal entry of pSN afferent collaterals), suggesting these could represent cutaneous low threshold mechanoreceptive afferents (Supplementary Fig. 2g). To examine this in more detail, we investigated innervation of tdT$^+$ axons in the glabrous and hairy skin of the mouse fore- and hindpaws. These analyses revealed that, in addition to muscle afferents, two types of skin projecting afferents appear to express both PV and Rx3 at some point during

their development. The first of these corresponds to Merkel cell afferents (characterized by their association with Troma1[+] Merkel cells) (Fig. 1e and Supplementary Fig. 3b), which were previously known to express Rx3, but not PV[27]. Intriguingly, we only observed tdT[+] Merkel cell afferents in forelimb skin and not in the skin of the back (Supplementary Fig. 3c), possibly suggesting that their (transient) expression of PV may depend on regional restricted regulatory control. The second class of cutaneous tdT[+] afferents we detected appears to innervate the nail (Fig. 1f and Supplementary Fig. 3d). The terminals of these neurons resemble slowly-adapting Ruffini endings that have previously been observed in the nailbed in human[7,33]. In contrast, tdT expression was not observed in longitudinal lanceolate endings or in afferents that associate with Meissner or Pacinian corpuscles (Supplementary Fig. 3e–i). Together these data demonstrate that use of the intersectional *PVRx3:tdT* reporter provides an efficient and nearly exclusive means to manipulate or isolate proprioceptive muscle afferents.

**Single cell RNA sequencing of proprioceptors reveals molecularly distinct subtypes**. The ability to label pSNs in DRG provides a means to significantly enrich these neurons for single cell transcriptome analysis without losing sequencing power to unrelated cell types. Considering the dynamic nature of proprioceptor subtype development[26,28], we chose to perform our initial analysis at an adult stage when the sensory system is fully mature. In addition, given that we expected the transcriptional differences between proprioceptors to be fairly limited (compared to differences between proprioceptors and nociceptors), we favored a high-depth method to detect mid- and low-expressed genes, as opposed to profiling more cells at lower depth[34]. For this reason, we selected a plate-based sequencing platform for all our transcriptome analyses—a choice we believe was justified based on a post-hoc analysis of sequencing data obtained from adult proprioceptors sequenced through plate- and droplet-based sequencing platforms (Supplementary Fig. 4a, b). To enable an unbiased molecular analysis of proprioceptor transcriptomes, DRG from all segmental levels were obtained from adult (≥p56) *PVRx3:tdT* mice. DRG were dissociated and single tdT[+] neurons were purified through fluorescent activated cell sorting (FACS), followed by plate sequencing (Fig. 2a). Neurons (480 in total) were sampled from three different experiments and derived from animals of either sex (totaling four males and two females). Cells with low gene detection (<2000 genes; 30 cells in total) or with significant contamination from attached satellite cells were eliminated by filtering for the satellite/Schwann cell markers Apoe and Mpz (cells with >10% of the Apoe/Mpz mean transcript level were removed from downstream analysis; 242 cells in total) (Supplementary Fig. 4c). After filtering out these contaminated cells we detected a mean of 6688 genes and 44,511 Unique Molecular Identifiers (a proxy for transcripts) per neuron for the remaining neurons (Supplementary Fig. 4d, e). With the exception of differences in global reads, we uncovered no qualitative differences between plates, and no batch correction was necessary.

To determine any underlying structure in our proprioceptor population we employed an iterative clustering method to identify putative groupings of cells with similar transcriptional profile (see "Methods"). These analyses resulted in the identification of five major clusters (Fig. 2b and Supplementary Fig. 4f). Each of these clusters shows robust expression of canonical muscle proprioceptor markers, including *PV*, *Rx3*, *Ntrk3*, *Etv1* and *Whirlin* (Fig. 2c and Supplementary Fig. 4f). We also identified five minor clusters (comprising fewer than 15 cells), which we omitted from downstream analysis. One of these minor

clusters does not express the pan proprioceptor marker *Whirlin*[35], suggesting that these neurons may correspond to some of the non-muscle afferent contaminants such as the Merkel cell afferents or Ruffini-like endings (Fig. 1e, f and Supplementary Fig. 3b, d).

Considering that muscle proprioceptors are classified as either MS (group Ia, group II) or GTO afferents (group Ib), we anticipated that our transcriptome analysis would reveal two, or possibly three, molecularly distinct clusters. The discovery of five clusters led us to consider if other features of the proprioceptor phenotype (e.g. regional identity) could contribute to the clustering. To examine this we determined the distribution of the expression of several known regional or muscle type molecules across the five clusters. With the exception of *Cadherin 13* (*Cdh13*) and to a lesser degree, *Cortactin 1* (*Crtac1*), we find that many regional restricted molecules (e.g. Hox transcription factors) or transcripts associated with either dorsal or ventral muscle targets (e.g. *Sema5a*, *Vstm2b*, respectively) are randomly dispersed throughout our five main clusters[36,37] (Supplementary Fig. 4f). Expression of *Cdh13* and *Crtac1* is biased to neonate proprioceptors that innervate dorsal or ventral limb muscle targets, respectively, but it remains unknown if this patterns is preserved in the adult[37]. These data suggest that the clustering revealed through our bioinformatics analysis is not driven by regional proprioceptor identity, but we cannot exclude the possibility that some clusters may reflect some regional bias.

**Validation of identified pSN clusters**. To investigate the biological relevance of the distinct pSN clusters, we first examined the features underlying the cluster distinctions. To do so, we performed an unbiased differential expression (DE) analysis between the neurons that comprise individual clusters (see methods). These analyses revealed molecules that are present in either single or subsets of pSN clusters (Fig. 2g). Interestingly, while many previously identified neonate proprioceptor subset markers show widespread expression across all five pSN clusters, the expression of three of these molecules appear biased towards certain clusters (Fig. 2c–g and Supplementary Fig. 4g–j)[28]. For example, expression of *Heart development protein with EGF like domains 1* (*Heg1*) is mainly associated with cluster 1, 2, and 3 neurons, while *Protocadherin 8* (*Pcdh8*) is prominently expressed in cluster 5 neurons (and at lower levels and frequency in clusters 2, 3, and 4) (Fig. 2d, f, g). In addition, *Neurexophilin* (*Nxph1*) shows expression in clusters 2, 3 and 4, with little or no expression in clusters 1 and 5 (Fig. 2e, g). Moreover, we find that the pattern of *Heg1*, *Pcdh8*, and *Nxph1* expression within the five bioinformatics-defined pSN clusters mirrors the combinatorial profile of these markers in adult DRG (Fig. 2g–j and Supplementary Fig. 4g). These data provide an indication that our pSN clusters represent distinct proprioceptor types observed in vivo.

We next used RNA in situ hybridization (RNA scope) to assess the DRG expression pattern for several identified cluster-restricted molecules not previously associated with adult proprioceptor subtypes (Fig. 3a). Specifically, we examined (1) what proportion of tdT[+] pSNs express these markers, and (2) the extent of overlap with molecules that define other pSN classes. For these experiments, we first examined the expression of the cluster 1 markers *Hpse*, *Colq*, and *Agpat4* in adult *PVRx3:tdT* DRG (Fig. 3b–g). Consistent with the proportion of cluster 1 neurons identified using bioinformatics approaches (51 out of 166), we find that these markers localize to ~20–40% of all tdT[+] neurons (Fig. 3c, e, g). We also observed that *Hpse*, *Colq*, and *Agpat4* are generally co-expressed, but excluded from neurons with high levels of *Pcdh8* expression—a marker of cluster 5 pSNs (Fig. 3b, d, f and Supplementary Fig. 5d, f, g). In contrast,

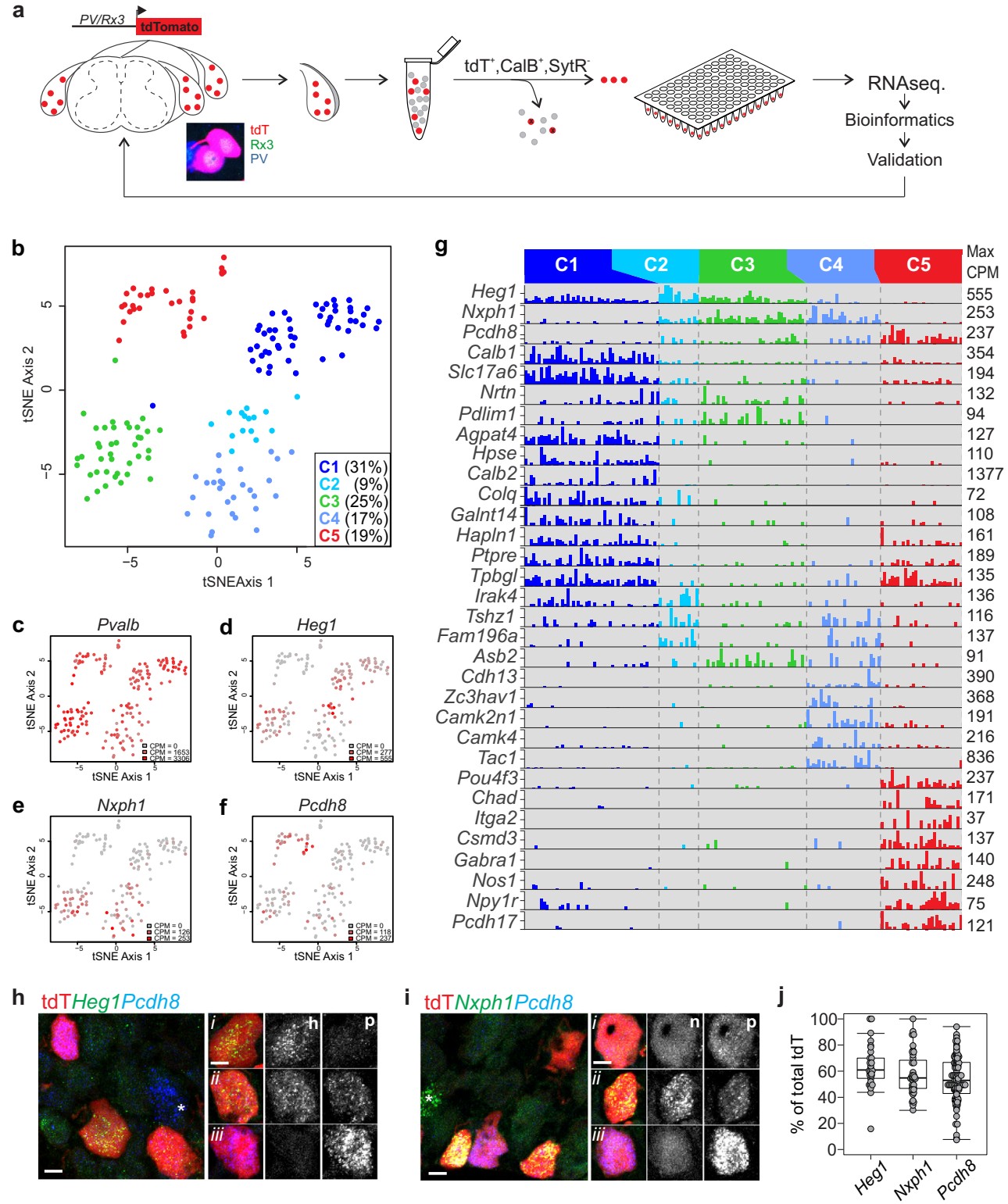

expression of the cluster 5 markers *Itga2*, *Chad*, and *Pcdh17* generally coincides with high levels of *Pcdh8* expression (as well as with each other) (Fig. 3h–k and Supplementary Fig. 5a, b, e). Expression of *Itga2*, *Chad*, and *Pcdh17* is observed in ~10–20% of all tdT[+] neurons, which is also in agreement with the proportion of cluster 5 neurons identified through our analysis (31 out of 166) (Fig. 3i, k and Supplementary Fig. 5b). We also considered if any of these cluster-selective transcripts correspond to afferents that have different intra limb distributions. To do so, we

examined the relative proportion of "marker[+]" pSNs in rostral lumbar DRG (L1-3; supplying body wall and proximal limb muscle targets) and caudal lumbar DRG (L4-5; supplying distal limb muscle targets). For each of the cluster 1 or 5 transcripts we examined, we find that their distribution is equivalent in rostral and caudal lumbar ganglia (Fig. 3c, e, g, i, k and Supplementary Fig. 5b). We next examined the expression of cluster 2–4 markers *Cdh13* (cluster 4) and *Fam196a* (cluster 2 and 4). Similar to cluster 1 and 5 transcripts, we find that both markers are

**Fig. 2 Single cell transcriptome analysis of PV+Rx3+ neurons. a** Schematic of single cell RNA sequencing strategy employed in experiments. DRG from adult (≥p56) *PVRx3:tdT* mice were collected and dissociated into single cell suspensions. tdT+ SNs were isolated using FACS. To ensure cell viability, neurons were also assessed for the presence of high Calcein Blue (CalB) and absence of Sytox red (SytR) labeling while sorting individual neurons into a single well of a 96-well plate containing lysis buffer (see "Methods" for details). **b** t-stochastic neighbor embedding (tSNE) plot of adult cells, colored by cluster membership. Clustering was done using a gene module and bootstrapping approach (with the hicat package—see "Methods"). **c-f** tSNE plots of cells colored by expression of four genes. In each plot, the color represents no expression (grey) to maximum counts per million (CPM) value over all the cells (red). **g** Barplots showing a subset of differentially expressed genes across the five main clusters. For each gene, the maximum CPM value over all cells is shown on the right side of the corresponding bar plot. Genes were selected for differential expression across multiple pairs of clusters with the added restriction of binary (on-off) expression, as far as possible, in at least two of the clusters. **h** Expression of *Heg1* and *Pcdh8* in tdT+ pSNs in lumbar DRG of adult (≥p56) *PVRx3:tdT* mice. Single neuron analysis (i–iii) of *Heg1* (h) and *Pcdh8* (p) expression exhibits three expression patterns across the five molecular identified clusters: cluster 1 tdT+ neurons expressing high transcript levels of *Heg1* but little *Pcdh8* (i), cluster 2-4 tdT+ neurons with comparable levels of *Heg1* and *Pcdh8* (ii), and cluster 5 tdT+ neurons with high *Pcdh8* but no *Heg1* (iii). Some *Pcdh8* expressing neurons fall outside of the tdT+ pSN population (marked by asterisk). **i** Expression of *Nxph1* and *Pcdh8* in tdT+ pSNs in lumbar DRG of adult *PVRx3:tdT* mice. Single neuron analysis (i–iii) of *Nxph1* (n) and *Pcdh8* (p) expression shows presumed (i) cluster 1 tdT+ neurons expressing no *Nxph1* but low levels of *Pcdh8* transcript, (ii) cluster 2–4 tdT+ neurons coexpressing *Nxph1* and *Pcdh8* (with variable levels of Pcdh8), and (iii) cluster 5 tdT+ neurons with no *Nxph1* but high *Pcdh8* transcript levels. Similar to *Pcdh8*, *Nxph1* is also expressed in a few non-tdT+ neurons (marked by asterisk). **j** Percentage of *Heg1+*, *Nxph1+*, and *Pcdh8+* tdT+ SNs in ≥p56 lumbar DRG of *PVRx3:tdT* animals. Data points indicate counts for a single DRG section and include both low and high expressing neurons. Mean percentage ± S.E.M. for *Heg1*: 63.5 ± 3.2%, *n* = 29 sections; for *Nxph1*: 57.8 ± 2.7%, *n* = 39 sections; for *Pcdh8*: 53.9 ± 1.9%, *n* = 94 sections. Similar data obtained from at least three (**h**, **i**) biological replicates. In boxplot (**j**), boxes indicate medians and 25th/75th percentiles, and whiskers extend to the furthest point less than 1.5 standard deviations from the mean. Scale 10 μm.

expressed in subsets of pSNs (Fig. 3l–o). In addition, within tdT+ neurons, *Cdh13* often overlaps with expression of the pan-cluster 2–4 marker *Nxph1+* (Fig. 3l and Supplementary Fig. 5f). Together these data establish that our molecularly defined pSN clusters largely consist of non-overlapping proprioceptor subsets.

**Molecularly defined pSN classes correspond to morphologically distinct MS and GTO afferents.** The observation that our molecularly defined pSN clusters correspond to different proprioceptor subsets prompted us to explore the biological relevance of these molecular distinctions. Specifically, we sought to determine if the neurons that comprise our different clusters align with any of the three known PSN subtypes: groups Ia and II MS afferents, or group Ib GTO afferents. We took advantage of the observation that among the various class-specific transcripts we identified a few markers for which validated immunological or genetic reagents are available. For class 1 neurons these transcripts are *Calbindin 2* (*Calb2*; encoding Calretinin), *Calbindin 1* (*Calb1*; encoding Calbindin), and the *vesicular glutamate transporter 2* (*Slc16a7*; encoding the vesicular glutamate transporter vGlut2); for class 4 neurons *Tachykinin 1* (*Tac1*; encoding Neurokinin A and Substance P); and for class 5 neurons, the transcription factor *Pou4f3* (encoding Brn3c) (Fig. 4a–f). To test if the expression of these molecules is restricted to either MS or GTO afferent proprioceptors, we examined their expression in adult muscle tissue, either directly (using specific antibodies) or indirectly (using genetic reporters) (Fig. 4g–k).

We first focused our analysis on the class 1 molecule Calretinin (CR). While expression of this marker is also observed in other DRG sensory neurons, we find that within muscle proprioceptors expression of CR is nearly exclusively restricted to afferents that innervate muscle spindles (276/322 MSs; 1/102 GTOs) (Fig. 4g, Supplementary Fig. 6a–c, and Supplementary Table 1). Moreover, CR+ afferents exhibit a highly regular spiral morphology and often appear to innervate the equatorial region of the spindle. In contrast, afferents that innervate the polar contractile ends of the spindle are devoid of CR (Fig. 4g and Supplementary Fig. 6a). Similarly, when using a *Calb2:Cre* allele in the context of the Cre-dependent *Mapt:lxp-STOP-lxp:GFP* reporter (hereafter *Calb2:GFP*)[30,38], MSs are typically innervated by a single vGlut1+GFP+ terminal which is most frequently positioned in the equatorial region of the spindle (Fig. 4h and Supplementary Fig. 6e–g). Based on these observations we conclude that, within muscle

proprioceptors, expression of CR marks group Ia MS afferents. Interestingly, while CR+ afferents are found in the vast majority of spindles we analyzed (85.7%), a small number of spindles appears to lack a CR+ afferent (Supplementary Fig. 6a, c and Supplementary Table 1). It is possible that this mosaic CR expression reflects the technical difficulties associated with immunological analysis of whole mount adult muscle, given that muscle size, the density of connective tissue, or the spindle capsule may prevent adequate antibody penetration. Alternatively, these differences may indicate differences in CR+ expression levels across different group Ia afferents. Consistent with the latter possibility, labeling of CR+ afferents using the *Calb2:Cre* reporter is efficient for Ia afferents with high CR expression levels (e.g. Extensor Digitorum Longus [EDL] or Extensor Carpi Radialis [ECR] muscle), but inefficient for Ia afferents with lower CR+ expression levels (e.g. Gluteus, Soleus, Plantaris) (Supplementary Table 1). In only one instance (out of 102 GTO's examined) did we observe a CR+ GTO afferent (Supplementary Table 1). Similar to CR, we find that expression of vGlut2 and Calbindin 1 (CB1) is most frequently associated with MS afferents that innervate the equatorial domain of the spindle (Fig. 4g, Supplementary Fig. 6d, and Supplementary Table 2). Together these data indicate that cluster 1 neurons, marked by expression of CR, vGlut2, and CB1, morphologically resemble group Ia MS afferents (Fig. 4l).

We next assessed the afferent terminals of cluster 5 Pou4f3+ neurons. In DRG, expression of Pou4f3 is observed in a relatively large proportion of neurons, yet labels just up to 25% of *PvRx3:tdT* neurons (Fig. 4c, d). To test if Pou4f3+tdT+ neurons constitute a specific functional subtype of proprioceptors, we used a Cre-conditional *Pou4f3* mutant and chromogenic reporter (*Pou4f3KOAP*) in the context of a *PV:Cre* allele[30,39]. When trans heterozygous for both alleles, *PV:Cre* mediates excision of one *Pou4f3* allele and simultaneously activates expression of Alkaline Phosphatase (AP) (which remains under the control of endogenous *Pou4f3* promoter elements) (*PVPou4f3:AP+*). Consistent with previous observations that most DRG neurons express *Pou4f3* at early embryonic stages[26], initially all pSNs (i.e. both MS and GTO afferents) are labeled by AP (Supplementary Fig. 7a–d). However, by p2, expression of AP is greatly diminished or absent in MS-innervating afferents but remains strongly expressed in GTO innervating afferents (Fig. 4i and Supplementary Fig. 7a, b). We observed *PVPou4f3:AP+* afferents

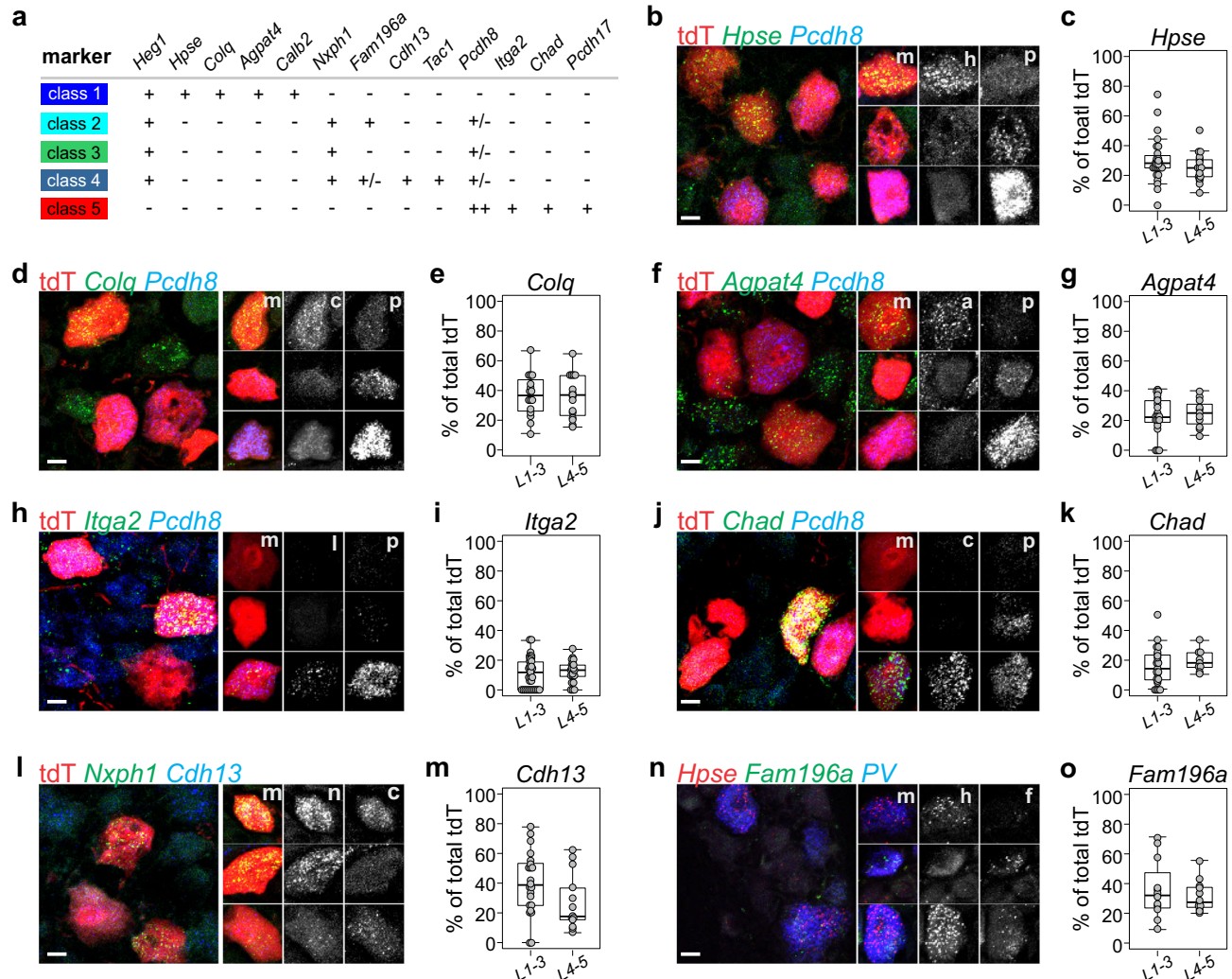

**Fig. 3 Validation of molecularly distinct proprioceptor subtypes. a** Summary of transcripts examined in validation assays. Transcript expression levels are indicated by "++" (high expression levels in most neurons), "+" (intermediate expression level in most neurons), "+/−" (lower expression levels in all or a subset of neurons), or "−" (no or nearly no expression). **b**–**g** Expression of cluster I transcripts *Hpse*, *Colq*, and *Agpat4* in relation to expression of the cluster 5 transcript *Pcdh8* in ≥p56 lumbar DRG of *PVRx3:tdT* mice. In **b**, **d**, **f**, images of individual neurons represent examples of observed transcript combinations other than those observed in the main image. High levels of *Hpse* (h) (in **b**), *Colq* (c) (in **d**), or *Agpat4* (a) (in **f**) transcript generally are mutually exclusive with high levels of *Pcdh8* (p) transcript (m indicates merged image). Percentage of *Hpse*+tdT+ neurons (**c**), *Colq*+tdT+ neurons (**e**), or *Agpat4*+tdT+ neurons (**g**) (of total tdT+) in ≥p56 rostral (L1-3) or caudal (L4-5) lumbar DRG of *PVRx3:tdT* animals. Mean percentage ± S.E.M. *Hpse* (L1-3): 30.1 ± 2.8%, n = 28 sections; *Hpse* (L4-5): 25.4 ± 2.2%, n = 19 sections; *Colq*(L1-3): 36.7 ± 3.5%, n = 16 sections; *Colq*(L4-5): 36.1 ± 3.6%, n = 16 sections; *Agpat4*(L1-3): 23.2 ± 2.8%, n = 21 sections; *Agpat4*(L4-5): 24.7 ± 2.8%, n = 11 sections. **h**–**k** Expression of the cluster 5 transcripts *Itga2*, *Chad*, and *Pcdh8* in ≥p56 lumbar DRG of *PVRx3:tdT* mice. In **h**, **j**, images of individual neurons represent examples of observed transcript combinations other than those observed in the main image. High transcript levels of *Itga2* (i) (in **h**) or *Chad* (c) (in **j**) generally overlap with high levels of *Pcdh8* (p) transcript expression (m indicates merged image). **i**, **k** Percentage of *Itga2*+tdT+ neurons (**i**) or *Chad*+tdT+ neurons (**j**) (of total tdT+) in ≥p56 rostral (L1-3) or caudal (L4-5) lumbar DRG of *PVRx3:tdT* animals. Mean percentage ± S.E.M. *Itga2*(L1-3): 11.5 ± 1.7%, n = 34 sections; *Itga2*(L4-5): 12.4 ± 1.6%, n = 21 sections; *Chad*(L1-3): 15.9 ± 2.6%, n = 23 sections; *Chad*(L4-5): 19.5 ± 2.4%, n = 9 sections. **l** Expression of cluster 2–4 transcript *Nxph1* and cluster 4 transcript *Cdh13* in ≥p56 *PVRx3:tdT* DRG. Images of individual neurons represent examples of observed transcript combinations other than those observed in the main image. Single neuron analysis confirms that a subset of *Nxph1* (n) neurons co-express *Cdh13* (c; m indicates merged image). **m** Percentage of *Cdh13*+tdT+ neurons (of total tdT+) in adult *PVRx3:tdT* DRG. Mean percentage ± S.E.M. *Cdh13*(L1-3): 39.6 ± 4.5%, n = 22 sections; *Cdh13*(L4-5): 27.3 ± 5.3%, n = 13 sections. **n** Expression analysis of *Hpse* (cluster 1), *Fam196a* (cluster 2 and 4), and *PV* in adult wild type DRG indicates the presence of cluster 1 *Hspe*+*Fam196a*off, and cluster 2 or 4 *Hspe*off*Fam196a*+ *PV* neurons. Images of individual neurons represent examples of observed transcript combinations other than those observed in the main image. A few cluster 1 neurons coexpress *Hspe* and *Fam196a* (see also Fig. 2g). **o** Percentage of *Fam196a*+tdT+ neurons (of total tdT+) in adult *PVRx3:tdT* DRG. Mean percentage ± S.E.M. *Fam196a*(L1-3): 35.9 ± 5.7%, n = 12 sections; *Fam196a*(L4-5): 31.4 ± 3.0%, n = 12 sections. Similar data obtained from at least three (**b**, **d**, **f**, **h**, **j**, **l**, **n**) biological replicates. In boxplots (**c**, **e**, **g**, **i**, **k**, **m**, **o**), boxes indicate medians and 25th/75th percentiles, and whiskers extend to the furthest point <1.5 standard deviations from the mean. Scale 10 µm.

in all muscles we analyzed (including Gluteus, Soleus, EDL, Plantaris, Gastrocnemius, axial muscle) in a pattern that is consistent with GTO afferents in these muscle targets (Supplementary Fig. 7b). These data imply that within pSNs, expression of *Pou4f3* is selective for GTO afferents, and consequently, that cluster 5 neurons correspond to group Ib proprioceptors (Fig. 4l). Consistent with these findings, we also observed Pcdh8 expression in GTO afferents but not in MS-innervating afferents (Fig. 4j).

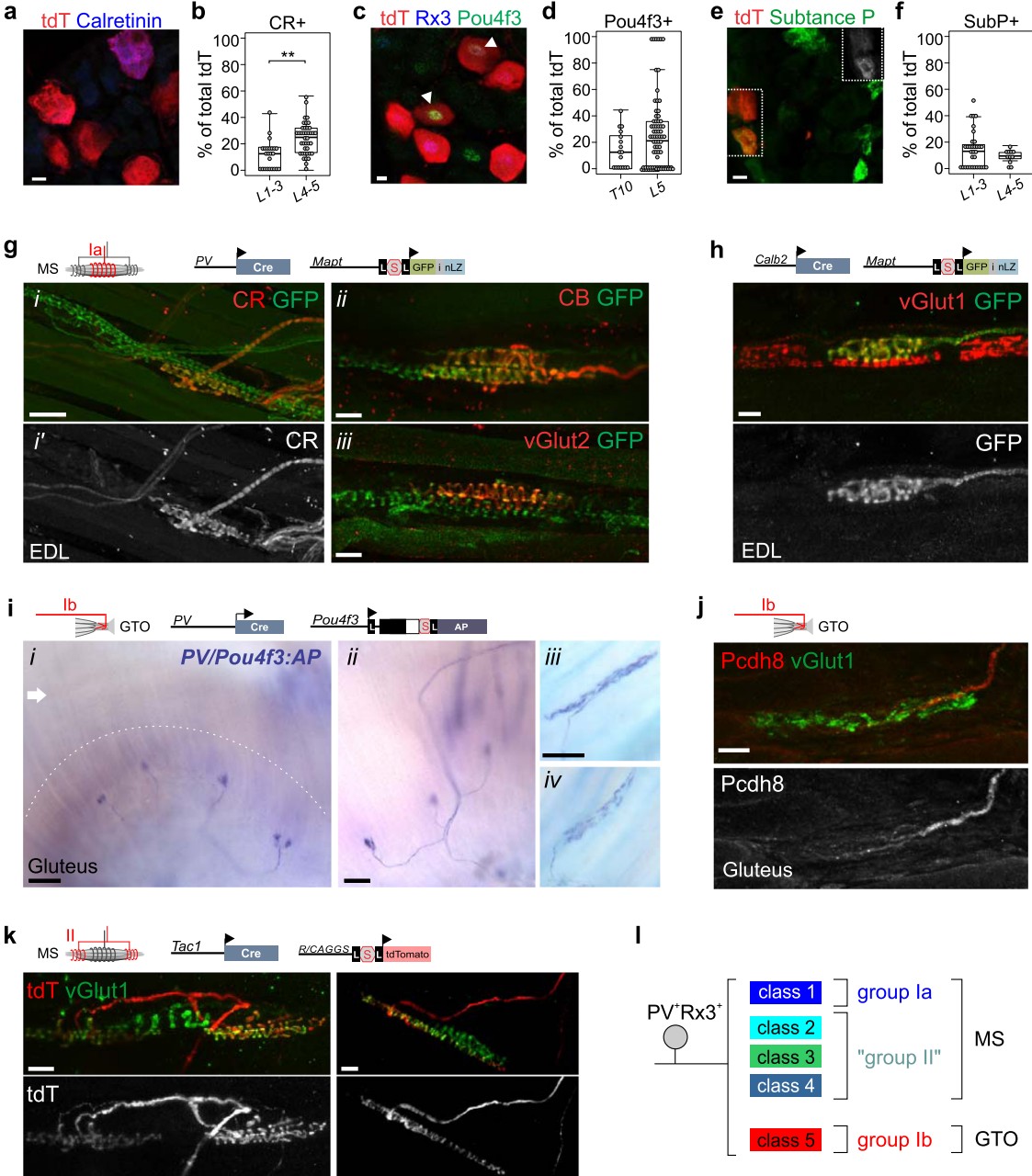

**Fig. 4 Molecular distinct pSN clusters segregate with MS or GTO afferents. a** Expression of Calretinin and tdT in ≥p56 lumbar DRG of *PVRx3:tdT* mice.
**b** Percentage of CR+tdT+ neurons (of total tdT+) in adult *PVRx3:tdT* DRG. Mean percentage ± S.E.M. CR(L1-3): 11.6 ± 2.3%, $n = 24$ sections; CR(L4-5):
23.9 ± 1.9%, $n = 40$ sections, $p < 0.001$ (two-sided Mann–Whitney *U* test). **c** Expression of Pou4f3, Runx3, and tdT in ≥p56 lumbar DRG of *PVRx3:tdT* mice.
Arrowheads indicate Pou4f3+Rx3+tdT+ neurons. **d** Percentage of Pou4f3+tdT+ neurons (of total tdT+) in adult *PVRx3:tdT* DRG. Mean percentage ± S.E.M.
Pou4f3(T10): 13.9 ± 3.6%, $n = 17$ sections; Pou4f3(L5): 25 ± 3.2%, $n = 78$ sections. **e** Expression of Substance P and tdT in ≥p56 lumbar DRG of *PVRx3:tdT*
mice. Boxed area is also shown in top right corner to illustrate expression of Substance P. **f** Percentage of Sub. P+tdT+ neurons (of total tdT+) in adult
*PVRx3:tdT* DRG. Mean percentage ± S.E.M. SubP(L1-3): 13.8 ± 2.5%, $n = 33$ sections; SubP(L4-5): 9.0 ± 1.6%, $n = 11$ sections. **g** Expression of Calretinin
(CR) (i, i'), Calbindin 1 (CB) (ii), and vGlut2 (iii) in GFP+ muscle spindle (MS) afferents of p12 (i, i') or adult (≥p56) (ii, iii) muscle of *PV:Cre; Mapt:lxp-
STOP-lxp:mGFP-iNLZ* (*PV:GFP*) mice. CR, CB, and vGlut2 expression is observed in GFP+ MS afferents that innervate the equatorial region of the muscle
spindle. **h** Expression of vGlut1 and GFP in a muscle spindle (MS) of p18 EDL muscle of *Calb2:Cre; Mapt:lxp-STOP-lxp:mGFP-iNLZ* (*Calb2:GFP*) mice. Similarly
as observed for Calretinin protein, in *Calb2:GFP* mice, expression of GFP labels MS afferents that innervate the equatorial region of the spindle. **i** Alkaline
phosphatase (AP) labeling of GTO afferents in gluteus muscle of *PV:Cre; Pou4f3:lxp-STOP-lxp:AP* (*PV/Pou4f3:AP*) at p2 (i, ii), and at ≥p56 (iii, iv) mice.
Dashed line in (i) indicates area of the myotentinous junction where GTOs reside. Arrow indicates muscle area where MSs are generally located. **j** Pcdh8
expression localizes to vGlut1+ GTO afferents in p12 gluteus muscle of wild type mice. **k** tdT+ MS afferents in EDL muscle of p20 *Tac1:Cre; Ai14:tdTomato*
(*Tac1:tdT*) animals. **l** Summary of molecularly distinct pSN subclasses and their association with either MS (group Ia, putative group II) or GTO (group Ib)
muscle receptors. In **g–l**, schematics indicate pSN subtype highlighted, and alleles used, in images below. Similar data obtained from at least three
(**a**, **c**, **e**, **g–i**), two (**k**), or one (**j**) biological replicates. In boxplots (**b**, **d**, **f**), boxes indicate medians and 25th/75th percentiles, and whiskers extend to the
furthest point <1.5 standard deviations from the mean. Scale 10 (**a**, **c**, **e**) 20 (**g**(ii), **g**(iii), **h**, **j**) or 50 (**g**(i)) μm.

The observation that cluster 1 neurons correspond to group Ia MS afferents and cluster 5 neurons to group Ib GTO afferents suggests that the remaining cluster 2–4 neurons, collectively marked by expression of *Nxph1*, may correspond to group II muscle afferents. Several lines of evidence support this idea. First, we and others previously showed that *Nxph1* expression in DRG is largely confined to proprioceptive sensory neurons[26,28]. Second, consistent with the notion that type II afferent neurons are smaller in caliber than groups Ia or Ib neurons[7,28], we showed that the cell bodies of *Nxph1*[+] neurons are smaller than the average *PV*[+] neuron cell body size (mean cell body diameter 24.5 ± 0.4 µm for *Nxph1*, $n = 88$ neurons; 27.6 ± 0.4 µm for *PV*, $n = 183$ neurons; $p < 0.001$, Student's *t* test). Third, based on morphological observations that MS are typically innervated by one group Ia and one to two group II afferents, the ratio of cluster 1 (group Ia) to cluster 2–4 (putative group IIs) we observed (1:1.53) aligns with the expected ratio for group Ia:group II afferents (1:1.5)[10]. To test if cluster 2–4 neurons represent group II afferents, we examined the expression of the cluster 4 marker *Tachykinin 1* (*Tac1*) in DRG and muscle. *Tac1* encodes Substance P and is typically associated with small diameter nociceptive neurons—not proprioceptors[40]. When examining Substance P expression in DRG of *PVRX3:tdT* mice, we confirmed that ~10–15% of tdT[+] neurons co-label with Substance P (Fig. 4e, f). To assess the peripheral sensory endings of Tac1[+] pSNs in muscle, we used a *Tac1:Cre* allele crossed to the Cre-dependent *Ai14:tdTomato* reporter (hereafter *Tac1:tdT*)[41,42]. These experiments confirmed that *Tac1:tdT*[+] afferents can innervate the polar ends of MSs, suggesting they correspond to group II afferents (Fig. 4k and Supplementary Fig. 8c). However, we also observed some *Tac1:tdT*[+] afferents that occupied the equatorial region of the spindle (indicating group Ia afferent identity), and noted numerous SubP[off]tdT[+] neurons in DRG (Supplementary Fig. 8b and Supplementary Table 3). These data indicate that some spindle innervating afferents (including group Ia afferents) may be labeled by the transient activation of *Tac1* expression in pSNs at earlier developmental stages. Together these data indicate that two of the transcriptionally distinct pSN clusters we identified, clusters 1 and 5, correspond to group Ia MS afferents and group Ib GTO afferents, respectively. By inference, while awaiting more definitive analyses, we postulate that the remaining identified molecular clusters (clusters 2–4) may correspond to group II MS afferents.

**Intrinsic and circuit properties of MS and GTO afferents revealed by differential expression of ion channels and neurotransmitter receptors.** The ability to correlate our molecularly defined pSN classes with morphologically distinct groups Ia, II, and Ib muscle afferents offers an opportunity to examine these pSN subtypes with respect to their subtype-selective expression of transcription factors, receptor molecules, and ion channels (Supplementary Fig. 9). (An interactive web application to search for individual genes in proprioceptor subtypes is available at https://vmenon.shinyapps.io/proprioceptors_scrnaseq). GO analysis demonstrates that the gene expression differences between the pSN clusters relate mostly ion channel function or synaptic membrane proteins (Supplementary Fig. 9a). However, a differential analysis of all voltage-gated sodium (Nav) or calcium (Cav) channels across the different proprioceptor classes revealed little variation in the expression of these channel subunits (Supplementary Fig. 10a, c). This finding may reflect the fact that these channels are a common feature of the general proprioceptor identity. One notable exception is *Scn7a*, which encodes the atypical $Na_x$ channel (also known as Nav2.3, NaG), and appears selectively enriched in group Ib (cluster 5) and presumptive group II (cluster 4) afferents (Supplementary Fig. 10c)[43]. In contrast to the relative uniform expression of Nav and Cav channels, we

noted a much larger diversity with respect to voltage-gated potassium (Kv) channels and their auxiliary subunits (Fig. 5a and Supplementary Fig. 10b). This diversity was apparent for individual pSN clusters but also between group I afferents (Ia and Ib) and presumed group II afferents. For instance, the A-type current modulator Kchip1 (*Kcnip1*) is upregulated in group I/cluster 1, 5 afferents relative to putative group II/cluster2–4 afferents, while Kchip2 (*Kcnip2*) is more prevalent in cluster2–4 afferents compared to group I afferents (Fig. 5a). Similarly, Kv7.3 (*Kcnq3*) is upregulated in group I afferents but shows little expression in cluster 2–4 neurons. Group I afferents also express higher levels of the Kv1.1 (*Kcna1*) and Kv1.2 (*Kcna2*) channels than prospective group II afferents (Fig. 5a).

Considering that potassium channels feature prominently in regulating neuronal firing properties, we wondered if the differences in the expression patterns of these channels would be reflected in the responses of pSNs to electrical stimuli. To address this question, we performed whole cell patch clamp recordings on isolated adult pSNs (identified using the *PVRx3:tdT* reporter) (Fig. 6a). Indeed, current injections revealed three firing patterns (Fig. 6b–f). Nearly half of the neurons exhibited rapid adaptation (RA) and fired a single action potential (AP), even at large current injections (51%; 22/43 neurons) (Fig. 6b–d). Other neurons (16.5%; 7/43 neurons) displayed a small burst of APs with increasing current injections (Fig. 6b, c, e), suggesting that they may represent an intermediate adapting (IA) subgroup. The remaining neurons exhibited a tonic firing pattern (32.5%; 14/43 neurons), with an average firing frequency of 53.95 ± 7.7 Hz (Fig. 6b, c, f). We note that these physiological analyses appear at odds with previous whole cell in vitro recordings of isolated PV[+] neurons (the majority of which were presumed to be pSNs). These prior studies detected either rapidly adapting (RA) or slowly adapting (tonic) neurons but not both[44,45]. Possible explanations for these discrepancies include differences in the temperature at which the recordings were performed, the duration of culture time following dissociation, and/or differences in the addition of growth factors during culture. Indeed, while we observed all three physiological neuronal classes when recordings were conducted at 37 °C, after 24 h of recovery at 37 °C and without addition of neurotrophins, our recordings at room temperature only revealed rapidly adapting (74% of neurons) and burst responses (26% of neurons).

We next determined if the observed pSN firing patterns correlate with any morphological or other electrical properties of the recorded neurons. While we did not observe major differences between soma sizes across the three physiologically identified neuronal types, single AP neurons were significantly smaller than burst neurons (mean ± S.E.M. was 762.9 ± 43.0 µm$^2$ for single AP, 1073 ± 98.9 µm$^2$ for burst AP, and 856.6 ± 65.4 µm$^2$ for tonic neurons) (Fig. 6g). In addition, we find that the resting membrane potential for tonic neurons was significantly more depolarized when compared to single AP neurons (Fig. 6h). Furthermore, while input resistance and voltage threshold were not statistically different among the three groups, capacitance was slightly lower in burst neurons compared to single AP neurons (Fig. 6j and Supplementary Fig. 11a, b), and rheobase was significantly lower in tonic neurons when compared to single AP neurons (Fig. 6i). We also find that the AP amplitude was statistically larger in tonic neurons compared to single AP neurons, but without any significant difference in AP half width (Supplementary Fig. 11c–e). Finally, repetitive stimulation revealed no difference between the various neuronal classes, and single AP, burst and tonic neurons all exhibited sustained 1:1 firing up to nearly 140 Hz (Supplementary Fig. 11f).

The voltage-gated potassium channels Kv1.1 and Kv1.2 were previously shown to influence adaptive response properties in other classes of low threshold mechanoreceptive sensory

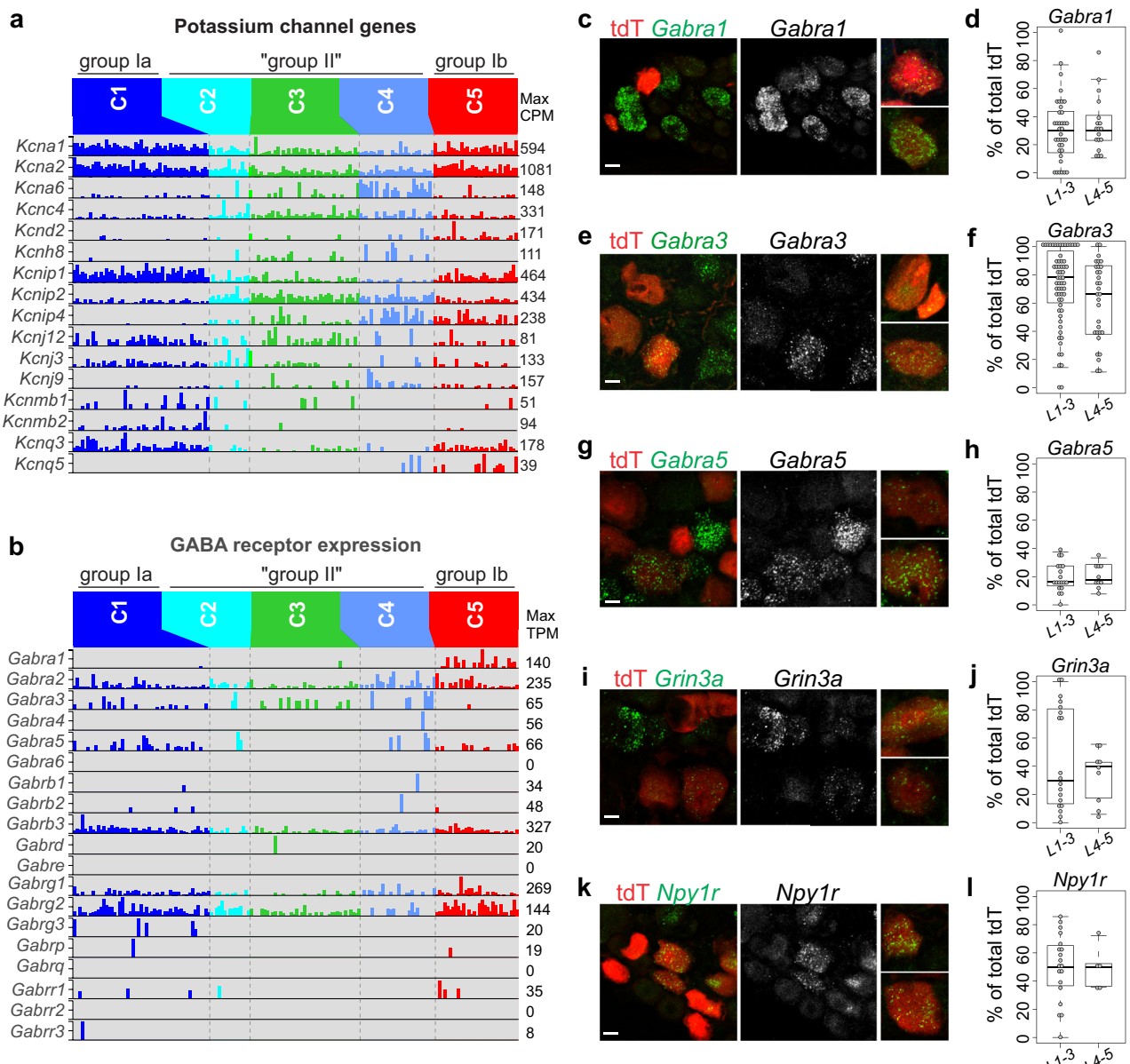

**Fig. 5 Circuit features of MS and GTO afferent sensory subtypes.** Bar graphs showing expression (in CPM) of annotated $K_v$ channel and auxiliary subunit genes (**a**), and GABA$_A$ receptor subunits (**b**) across the five putative cell types identified from the clustering of single-cell RNAseq data. **c–h** Distribution patterns of the GABA$_A$ receptor units *Gabra1* (**c**), *Gabra3* (**e**), and *Gabra5* (**g**) in tdT$^+$ neurons in adult (≥p56) *PVRx3:tdT* mice. Images of individual neurons represent examples of observed *Gabra$^+$*tdT$^+$ neurons other than those observed in the main image. Percentage of *Gabra1$^+$*tdT$^+$ (**d**), *Gabra3$^+$*tdT$^+$ (**f**), and *Gabra5$^+$*tdT$^+$ (**h**) (of all tdT$^+$ neurons) in p56 mice, at rostral (L1-3) and caudal (L4-5) lumbar levels. Mean percentage ± S.E.M. *Gabra1*(L1-3): 30.6 ± 3.8%, $n = 37$ sections, *Gabra1*(L4-5): 34.2 ± 4.8%, $n = 18$ sections; *Gabra3*(L1-3): 73.1 ± 3.7%, $n = 63$ sections, *Gabra3*(L4-5): 61.4 ± 4.9%, $n = 31$ sections; *Gabra5* (L1-3): 19.3 ± 2.4%, $n = 19$ sections, *Gabra5*(L4-5): 20.5 ± 2.7%, $n = 10$ sections. **i** *Grin3a* receptor expression in lumbar DRG of ≥p56 *PVRx3:tdT* mice. *Grin3a* transcript is detected in a subset of tdT$^+$ neurons. Images of individual neurons represent examples of observed *Grin3a$^+$*tdT$^+$ neurons other than those observed in the main image. **j** Percentage of *Grin3a$^+$*tdT$^+$ neurons (of all tdT$^+$ neurons) in lumbar DRG of ≥p56 *PVRX3:tdT* mice. Mean percentage ± S.E.M. *Grin3a*(L1-3): 45.9 ± 8.4%, $n = 19$ sections, *Grin3a*(L4-5): 33.3 ± 6.4%, $n = 9$ sections. **k** Expression of *Npy1r* transcript in tdT$^+$ neurons in lumbar DRG of ≥p56 *PVRx3:tdT* mice. Images of individual neurons represent examples of observed *Npy1r$^+$*tdT$^+$ neurons other than those observed in the main image. **l** Percentage of *Npy1r$^+$*tdT$^+$ neurons (of all tdT$^+$) in lumbar ganglia. Mean percentage ± S.E.M. *Npy1r*(L1-3): 49.9 ± 5.4%, $n = 19$ sections, *Npy1r*(L4-5): 49.3 ± 6.7%, $n = 5$ sections. Similar data obtained from at least three (**c**, **e**, **g**, **i**, **k**) biological replicates. In boxplots (**d**, **f**, **h**, **j**, **l**), boxes indicate medians and 25th/75th percentiles, and whiskers extend to the furthest point less than 1.5 standard deviations from the mean. Scale 10 μm.

neurons[45]. Therefore, we next asked if the observed proprioceptor firing responses may correlate with the aforementioned differences in the expression levels of Kv1.1 and Kv1.2 between groups Ia and Ib afferents (clusters 1 and 5; high levels of *Kcna1* and *Kcna2*) and presumptive group II afferents (clusters 2–4; lower *Kcna1/2* levels). To test this, we explored the effect of alpha-

Dendrotoxin (DTXα), an established inhibitor of these potassium channels[45,46]. We find that application of DTXα (20 nM) consistently reverts pSN phasic responses (Single AP and Burst neurons) to a tonic firing pattern in response to current injection (Fig. 6k–n and Supplementary Fig. 11g–i). In contrast, tonic responses are largely unaffected by DTXα, with the exception of a

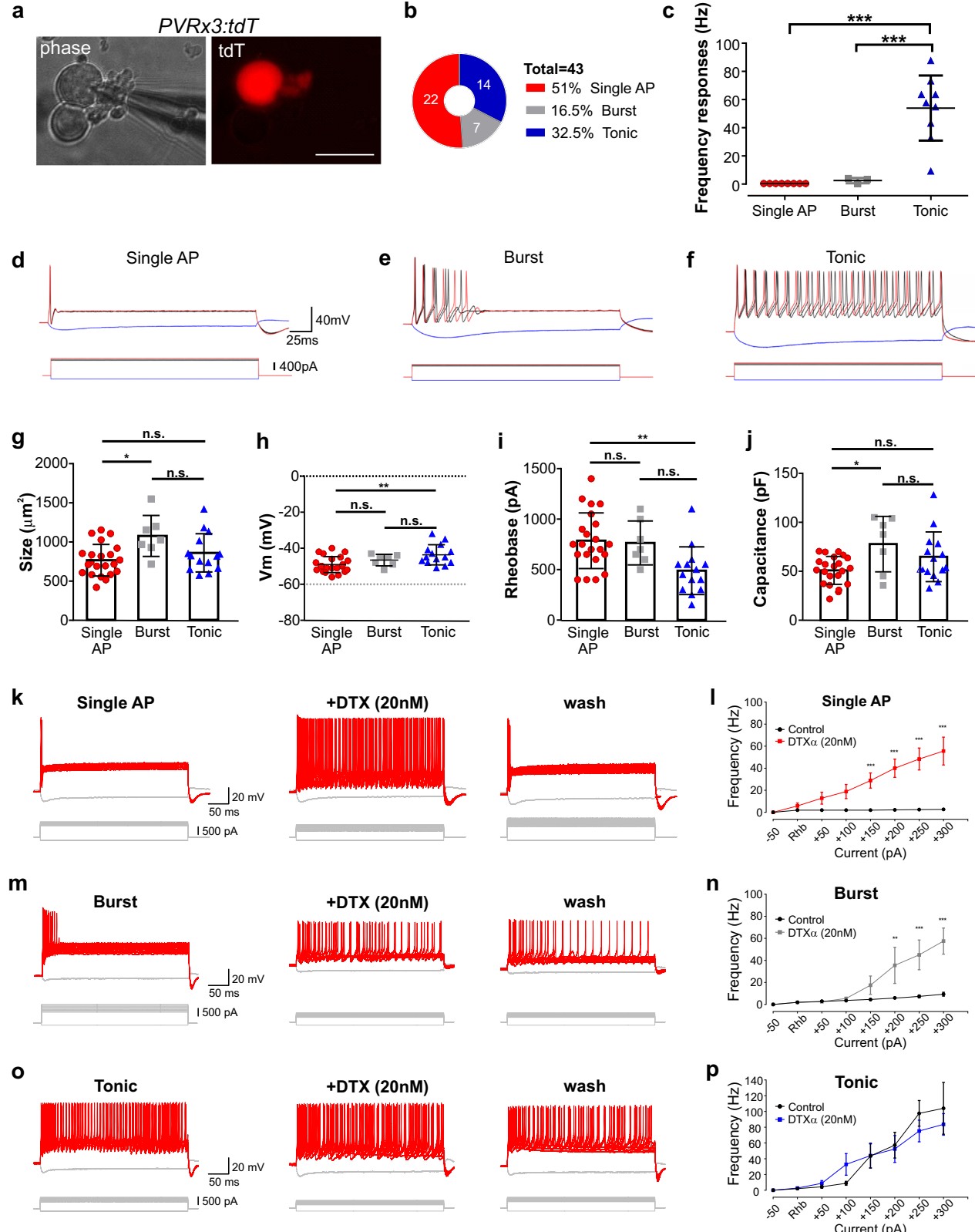

reduction in rheobase (Fig. 6o, p and Supplementary Fig. 11g). These observations indicate that the differential expression patterns of Kv ion channel subunits contribute to the unique features of group I and group II proprioceptors. By extension, these data support the idea that the recorded tdT+ pSN neurons with dynamic (RA or IA) firing properties comprise groups Ia and/or Ib afferents.

Besides intrinsic group I and putative group II afferent-specific physiological features, our pSN transcriptome profiling offers insight into how MS and GTO afferent subtypes may operate at the circuit level. For example, we find that the five identified pSN classes exhibit different patterns of GABA$_A$ receptors: while all afferents express *Gabra2*, group Ia MS afferents co-express *Gabra3* and *Gabra5*, presumptive group II MS afferents co-express

**Fig. 6 Electrophysiological properties of muscle proprioceptors. a–j** Physiological properties of adult (>p28) dissociated tdT$^+$ neurons as assessed through in vitro patchclamp recordings. **a** Phase contrast images of adult DRG neurons in culture. A patch electrode is shown (left) on a PV$^+$ neuron expressing tdTomato (red; right). Scale 100 μm. **b** The firing patterns for DRG PV$^+$ neuron expressing tdTomato displayed in percentages. **c** The frequency response for the three different firing patterns in proprioceptive neurons ($n = 22$); color scheme as in **b**. Tonic firing proprioceptors exhibited significantly higher frequencies compared to proprioceptors in which injected current (200–400 pA supra-threshold current steps for 2.5 s) evoked a single AP or burst firing. Each dot represents a single neuron, bars represent mean ± S.D. ($p < 0.001$; One-way ANOVA; Tukey's post hoc test). n.s. not significant. **d** Superimposed traces of voltage (top) following current injection (bottom) for proprioceptors exhibiting a single action potential (AP) irrespective of the current injected. **e** Similar to **d**, for a proprioceptor exhibiting a short duration burst of action potentials. **f** Traces from a tonic firing proprioceptor. Bar graphs showing differences in soma size (**g**), resting membrane potential (**h**), Rheobase (**i**), and Capacitance (**j**) across proprioceptor neurons ($n = 43$). Data expressed as mean ± S.D.; $p$ values for size $p = 0.0112$; resting membrane potential (Vm) $p = 0.0065$; rheobase $p = 0.0046$; capacitance $p = 0.0117$ (one-way ANOVA; Tukey's post hoc test). n.s. not significant. **k–p** Superimposed traces of voltage responses (top) following current injection (bottom) for proprioceptors exhibiting a single action potential (AP; irrespective of the current injected) (**k**) ($n = 12$), burst firing (**m**) ($n = 5$), or tonic firing (**o**) ($n = 4$), before, during, and after wash-out of 20 nM DTXα. Frequency-current plots for Single AP (**l**), burst (**n**), or tonic (**p**) neurons in response to current steps (relative to rheobase) before or during application of DTXα. Data expressed as mean ± S.E.M.; $p$ values. **l** Single AP, current −50 pA ($p > 0.999$), rheobase ($p = 0.6007$), +50 pA ($p = 0.1404$), +100 ($p = 0.0526$), +150 pA ($p = 0.0006$), +200 pA ($p < 0.0001$), +250 pA ($p < 0.0001$), +300 pA ($p < 0.0001$). **n** Burst, current −50 pA ($p > 0.999$), rheobase ($p > 0.999$), +50 pA ($p = 0.9579$), +100 ($p = 0.8329$), +150 pA ($p = 0.1746$), +200 pA ($p = 0.0057$), +250 pA ($p = 0.0005$), +300 pA ($p < 0.0001$). **p** Tonic, current −50 pA ($p > 0.999$), rheobase ($p = 0.9624$), +50 pA ($p = 0.7957$), +100 ($p = 0.1611$), +150 pA ($p = 0.9812$), +200 pA ($p = 0.7776$), +250 pA ($p = 0.1904$), +300 pA ($p = 0.3005$) (two-tailed paired $t$-test).

*Gabra3*, and group Ib GTO afferents co-express *Gabra1* and *Gabra5* (Fig. 5b–h). Considering that proprioceptive muscle afferents are subject to central regulation through presynaptic inhibition[47], the differential expression of these *Gabra* subunits could offer a mechanism to regulate this central influence on MS and GTO afferents separately. In addition, we note that group Ib GTO afferents, but not groups Ia or II MS afferents, express appreciable mRNA levels of the G-protein coupled NPY receptor *Npy1r* and the NMDA receptor *Grin3a* (Fig. 5i–l and Supplementary Fig. 10d). Similar to *Gabra* receptor units, the differential expression of these molecules in pSN subsets may offer insight into proprioceptor engagement or inhibition within spinal circuits.

**Emergence of pSN subtype identity.** To reveal when proprioceptive afferent identities first emerge, or the molecular mechanism that underlies their development, we also performed single cell RNAseq at key time-points during proprioceptor development. These stages included e14.5, when proprioceptors innervate their nascent receptor targets; p0, when animals are born (and first begin to apply goal-directed movement); and p12, when animals are able to rear themselves and walk. Similar to adult proprioceptors, we used the intersectional *PVRx3:tdT* genetic reporter to isolate p0 and p12 pSNs (Fig. 7a). Since PV is expressed at low levels at early embryonic stages, to efficiently isolate e14.5 pSNs we used a *TrkC-tdTomato* allele (hereafter *TrkC:tdT*)[48], which labels all Rx3$^+$ neurons at this stage (including PV$^+$Rx3$^+$ pSNs and PV$^{off}$RX3$^+$ Merkel cells afferents) (Fig. 7a). For all age groups, neurons were obtained from at least two different animals, and similar to the adult dataset, neurons were assessed for quality and were filtered for satellite cell contaminants. The total number of neurons with sufficient coverage (>2000 genes detected) and low contamination (<10% of mean Apoe, Mpz contaminant transcript counts) was 169 for e14.5, 55 for p0, and 154 for p12 (Supplementary Fig. 12a–e). Cluster analysis of all datasets combined demonstrated a nearly complete segregation by developmental stage, suggesting that maturation-related transcripts dominate the clustering over any other potential proprioceptor markers (Fig. 7b). Unsupervised clustering of the individual e14.5, p0, p12 cellular transcriptomes was performed by applying the same iterative clustering method as described for adult neurons (see "Methods"). These analyses revealed four clusters for the e14.5 neurons, three clusters for p0, and five clusters for the p12 neurons (similar as for adult) (Fig. 7c–e). For the e14.5 data set however, two clusters (C3 and C4) show a near absence of expression of

*ETV1* and *PV*, and an increased level of *Maf* expression (Supplementary Fig. 12f, 13a). These data suggest that these two e14.5 clusters represent subsets of non-proprioceptive *TrkC:tdT*$^+$ neurons, including Merkel cell afferents[27]. Together these findings indicate that pSNs progressively become more diverse (from two to five clusters) during development.

To assess to what extent the genes that define pSN classes in the adult correlate with gene expression patterns at earlier developmental stages, we next performed a lineage analysis focusing on transcripts that identified MS or GTO proprioceptor subtypes in the adult. Genes with strong co-expression patterns in adult clusters have weaker associations at p12 and essentially no associations at earlier time points (with exception of a few GTO afferent markers at p0) (Fig. 7f). These data suggest that the expression patterns seen in adult pSN classes are not anticipated by the transcript patterns observed at earlier time points, but rather result from a dynamic interaction of lineage-defining genes. Consistent with these observations, we find that expression of CR is not observed in Rx3$^+$PV$^+$ pSNs at p0 (Fig. 7g, h). Instead, CR only approaches its adult pattern of expression at p12 (Supplementary Fig. 13b). Similarly, Substance P-expressing pSNs are rare at p0 and only reach their adult expression pattern at p12 (Fig. 7k, l and Supplementary Fig. 13c). In contrast, Pou4f3 is expressed in many, if not all, DRG neurons at early stages and becomes confined to a subset of pSNs only later (Fig. 7i, j and Supplementary Fig. 13a). These data suggest that the differential maintenance of Pou4f3 expression is one of the first molecular distinctions between MS and GTO afferents.

The observation that expression of *Pou4f3* may predict pSN GTO subtype identity between e14.5 and p0 led us to consider if other transcription factors may anticipate the MS–GTO lineage selection at this developmental stage. To address this question, we performed a differential expression analysis at e14.5 and p0, focusing on the set of transcription factors that show elevated levels of expression in at least one class at either age group (Supplementary Fig. 14). These analyses again demonstrate elevated levels of *Pou4f3* expression in the p0-C2 cluster (suggesting they represent GTO afferents), but show little distinction in *Pou4f3* expression between the two pSN clusters (C1 and C2) observed at 14.5. Similarly, while this analysis revealed other transcription factors that are differentially expressed between MS and GTO afferents at p0 (e.g., *Id1, Id2, Id3, Tcf15, Tcf19, Crip1, Zfp235, Tcf7l2*), with the exception of Zfp235, none of these transcription factors are differential expressed at e14.5 (Supplementary Fig. 14). Conversely, transcription factors that are

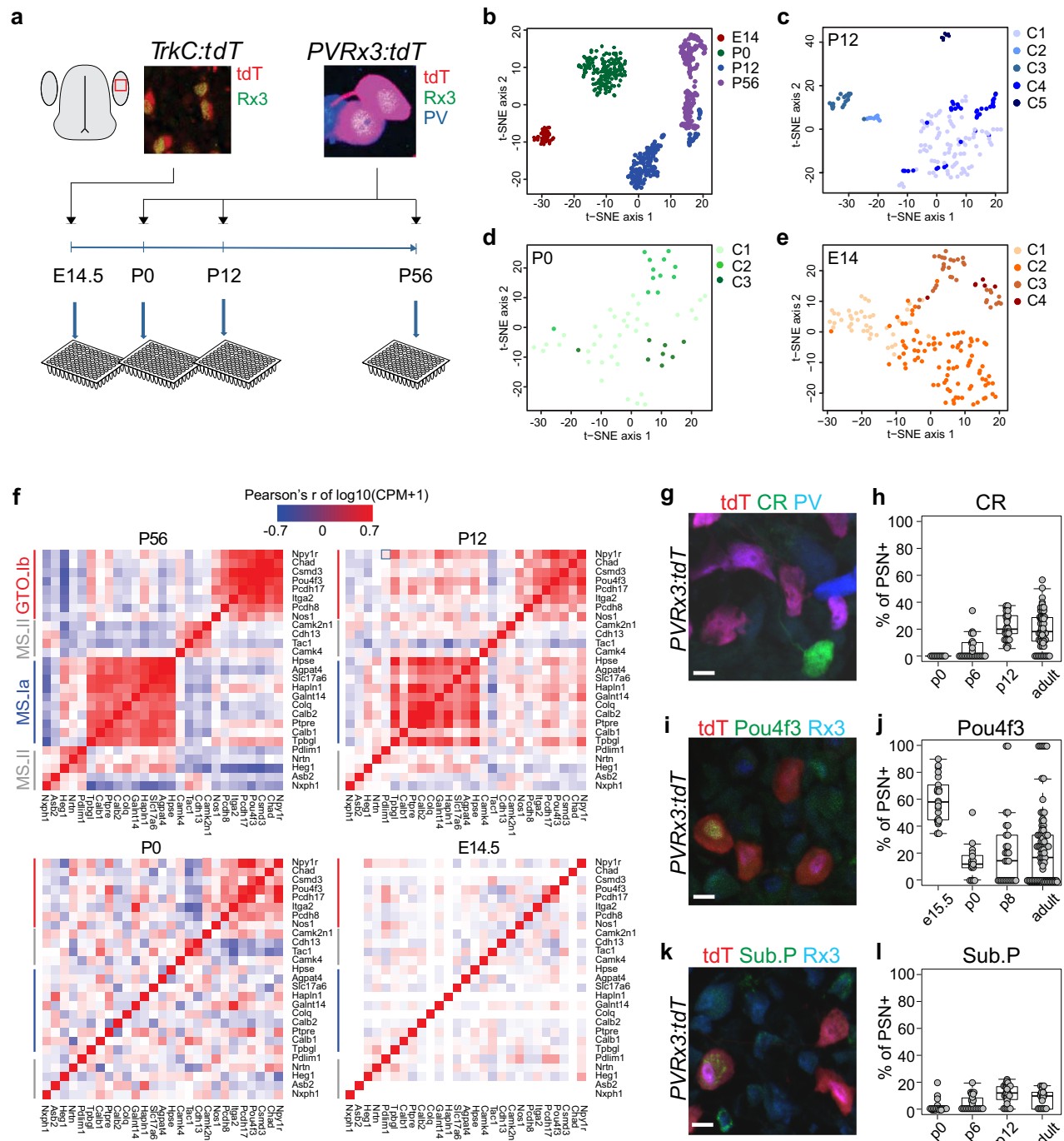

differentially expressed between the C1 and C2 groups at e14.5 (*Casz1, Cux2, Pou4f1, Spry2, Spry4*) are equally distributed across all three p0 clusters. Thus, while future sequencing efforts may further refine transcriptional insights into embryonic pSNs, our data reinforce the idea that proprioceptive muscle afferents remain uncommitted during their early trajectory towards their peripheral target[28], and that transcriptional MS/GTO subtype identities emerge over a prolonged developmental period that extends after the afferents innervate their nascent receptors.

## Discussion

Proprioceptive muscle feedback is relayed though muscle spindle and GTO sensory afferents, yet little is known about the diversity among these afferents beyond the broad distinctions between their receptor and central targets. Here, we used single cell RNA sequencing and transcriptome analysis of genetically identified adult, adolescent, neonatal, and embryonic muscle proprioceptors to provide long overdue insights into the molecular differences between these afferent neurons. While previous transcriptome studies have revealed general proprioceptor markers based on comparisons to other somatosensory modalities, our unbiased analyses selectively focused on proprioceptive muscle afferents. This enabled us to identify proprioceptor subtype-selective transcripts that otherwise might exhibit more widespread expression in DRG. Apart from uncovering MS and GTO afferent-selective markers, our transcriptome analysis reveals insights into the intrinsic physiological properties of proprioceptors and reinforces the idea that proprioceptor subtype identity may emerge though extrinsic signals.

**Fig. 7 Proprioceptor transcriptome profiling shows dynamic transcriptional changes across development. a** Experimental paradigm for proprioceptor transcriptome analysis across different developmental stages. Developmental stages include e14.5, p0, p12, and adult (≥p56). p0 and p12 neurons were isolated as described for the adult dataset. Due to the inefficient *PV:Cre*-mediated recombination at early developmental stages, e14.5 neurons were isolated using a *TrkC:tdTomato* reporter. As described for adult pSNs, viable tdT$^+$ pSNs neurons were isolated by FACS and directly deposited into 96-well plates. **b–e** tSNE plots at each time point profiled, with cells colored by cluster identity, using the same clustering technique (see "Methods") used to identify putative cell types in the adult in Fig. 2. **f** Coexpression patterns of genes with observed coexpression in the adult. For each time point, the heatmap represents pairwise gene–gene correlation values (Pearson's *r* using log-transformed CPM data). Because genes were selected based on co-expression in the adult (a subset of cluster-specific genes shown in Fig. 2g), the correlations are strongest in the adult. At p12, only the putative group Ib genes show strong correlations, whereas by p0 and e14, the overall correlations are substantially weaker. This suggests that general cell type-specific gene coexpression patterns in the adult differ from those in development. **g–l** Developmental expression patterns of Calretinin (CR), Pou4f3 and Substance P (Sub.P). Expression of CR (**g**), Pou4f3 (**i**), and Substance P (**k**) in p0 lumbar DRG of *PVRx3:tdT* mice. Percentage of CR$^+$ (**h**), Pou4f3$^+$ (**j**), or Sub. P$^+$ (**l**) pSNs (defined by *PVRx3:tdT*$^+$ or PV$^+$Rx3$^+$ neurons) at different developmental stages. Mean percentage ± S.E.M. for CR at p0: 0 ± 0%, n = 10 sections; at p6: 4.8 ± 1.6%, n = 26 sections; at p12: 22.5 ± 1.8%, n = 29 sections; at ≥p56: 19.4 ± 1.6%, n = 65 sections. Mean percentage ± S.E.M. for Pou4f3 at e15.5: 59.6 ± 3.5%, n = 22 sections; at p0: 13.5 ± 3.0%, n = 17 sections; at p6: 21.4 ± 5.2%, n = 28 sections; at ≥p56: 23.1 ± 2.8%, n = 95 sections. Mean percentage ± S.E.M. for Sub. P at p0: 2.0 ± 1.1%, n = 20 sections; at p6: 4.5 ± 1.2%, n = 25 sections; at p12: 11.3 ± 1.2%, n = 30 sections; at ≥p56: 8.8 ± 1.3%, n = 22 sections. Similar data obtained from at least three (**g**, **i**, **k**) biological replicates. In boxplots (**h**, **j**, **l**), boxes indicate medians and 25th/75th percentiles, and whiskers extend to the furthest point <1.5 standard deviations from the mean. Scale 10 µm.

Using transcriptome analysis of single proprioceptors, we identified molecular markers for the main pSN subclasses, including groups Ia MS afferents and group Ib GTO afferents. Specifically, among many other differentially expressed molecules, we demonstrate that expression of *Calretinin, Calbindin 1*, and *vGlut2* marks group Ia MS afferents, while expression of *Pcdh8* and *Pou4f3* is observed in group Ib GTO afferents. By inference, given that two of the molecular clusters we identified account for nearly all group Ia MS afferents and group Ib GTO afferents, respectively, our data also suggests that *Nxph1* may be a general marker for group II MS afferents. Although this assessment requires further verification using additional genetic reagents, these results reveal a larger molecular diversity among MS afferent subtypes than a priori anticipated. The functional significance of this afferent diversity remains unclear. Yet, it is interesting to note that in vivo electrophysiological recordings have revealed classes of MS afferents that do not fully fit the firing behaviors expected for either group Ia or group II afferents, but instead exhibited mixed Ia/II response properties with some group Ia's lacking vibration sensitivity and some group II's showing dynamic sensitivity[15]. Our data may suggest that the electrophysiological variability observed in vivo could in part have a molecular basis. We expect that future genetic strategies targeting individual MS afferent subtypes, beyond providing insight into their overall role in motor control, may reveal the biological relevance—if any—of this molecular subtype diversity in the organization of body and limb position sense.

A second surprising element of our studies concerns the identity of the transcripts that we found to define the different adult proprioceptor classes. Several of these molecules have not before been associated with a proprioceptor identity (*Hspe, Colq, Chad, Agpta4*), while other markers have typically been observed in sensory neurons other than proprioceptors (e.g. *vGlut2, Tac1*)[49,50]. The latter observation, in particular, suggests that identification of sensory neuron subtypes on the basis of one or two molecular markers is an unreliable strategy, and as similarly observed for other neuronal classes, requires 'multi-transcript authentication'[51,52]. Our data indicate that vGlut2, together with Calretinin and Calbindin, marks group Ia MS afferents. The expression of Calretinin and Calbindin group Ia MS afferents is not surprising given that both molecules are often observed to regulate Ca$^{2+}$-homeostasis in highly active neurons[53]. However, the observation that adult group Ia MS afferents also prominently express vGlut2, in addition to vGlut1, was unexpected, given that vGlut2 (unlike vGlut1) is barely detectable in Ia afferent central terminals[54,55]. Why group Ia afferents would require both vGlut1

and vGlut2 in the sensory ending remains as of yet unresolved. A similarly intriguing observation is the expression of *Tac1* in a subclass of presumed group II afferents. *Tac1* encodes the neuropeptides Neurokinin A and Substance P, and represents one of the hallmarks of a nociceptive sensory identity[40]. The expression of *Tac1* in group II MS afferents may hint at a direct role for muscle afferents in inflammatory muscle pain. Besides these MS afferent markers, the biological role for most MS or GTO afferent molecules here identified remains largely unknown, let alone their relevance to pSN function. Thus, a better understanding of the biological activities of these molecules in general will likely offer further insights into proprioceptor development and/or function.

Aside from molecular markers for proprioceptor subtypes, our transcriptome data offers valuable insights into the molecules that control the intrinsic electrical properties for individual proprioceptor subtypes. In contrast to voltage-gated sodium and calcium channel subunits, we detected a large degree of variability in the expression of voltage-gated potassium channels across the five pSN classes. These include differences between individual proprioceptor subtypes, as well as global differences between group I and presumed group II afferents. Indeed, whole cell patch clamp recordings of pSNs revealed three main firing patterns upon current injection: rapidly-, intermediate-, or slowly adapting. Based on prior observations, we postulate that rapidly-adapting (RA) and intermediate-adapting (IA) neurons may comprise group Ia or Ib afferents, while slowly-adapting (SA) tonic neurons may correspond to group II afferents[11,15,16]. While these afferent firing patterns may result from differences in several types of ion channels, we observed higher levels of Kv1 channel subunit expression in group I (clusters 1 and 5) afferents compared to presumed group II (clusters 2–4) afferents. This is of note because Kv1 subunits previously were implicated in regulating the firing properties of Aβ SA and Aβ RA afferents: afferents with lower Kv1 levels demonstrated tonic firing patterns, while those with higher levels adapted rapidly[45]. Consistent with these observations, we demonstrate that application of alpha-Dendrotoxin (DTXα; an inhibitor of Kv1.1/Kv1.2 channels) consistently reverts the phasic response to a tonic firing pattern in response to current injection. In contrast, tonic responses are largely unaffected by DTXα. These findings not only suggest that the differential expression of these potassium channels contributes to the unique features of proprioceptor firing properties, but also support the idea that group Ia and Ib afferents comprise afferents that exhibit dynamic firing properties. Nevertheless, a definitive link between the observed physiological features and pSN subtype identity will require direct recordings of molecularly identified pSNs.

The tonic firing we observed in some pSNs is consistent with observations that group IIs provide information regarding static muscle length and generally have little dynamic sensitivity[7,11,15]. The phasic (RA and IA) pSN responses we observed are seemingly at odds with in vivo observations that group Ia and group Ib afferents relay both static and dynamic information about muscle length and contraction, respectively, but are reminiscent of the firing patterns described for Merkel cell afferents[11,16,44,56]. Merkel cell afferents exhibit RA and SA firing upon Merkel cell stimulation in vivo, but only show a RA response to current injection in in vitro recordings. Thus, the SA responses observed for groups Ia and Ib afferents in vivo may be due to receptor-based modulation (similar as proposed for Merkel cell afferents), possibly through sustained glutamatergic activation at the sensory terminal[56,57]. Lastly, it is important to note that current injection experiments do not expose the full repertoire of electrical responses and future studies may yet reveal physiological distinctions between pSN afferents that reflect other differences in ion channel expression patterns.

Molecules that distinguish MS afferents from GTO afferents have thus far remained elusive. The inability to uncover such markers in previous studies may possibly be attributed to the long held conception that proprioceptor subtype identity is already established prior to pSN engagement with their peripheral receptors, thus targeting the search for such molecules to early developmental stages. Indeed, expression of Calretinin (CR) was previously noted in group Ia muscle afferents in the chicken and rat[58]. Yet our past studies consistently failed to observe CR in embryonic or neonatal mouse proprioceptors, leading us to conclude that CR presented a species-specific difference between chick, rat and mouse. We now confirm that CR is not expressed in embryonic or early postnatal proprioceptors and only reaches its mature pattern of expression by p12. Similarly, we find that expression of Tac1, a presumptive group II marker, only reaches its mature pattern of expression by p12. Both observations are supported by our longitudinal transcriptome analysis of e14.5, p0 and p12 proprioceptors and together imply that distinctions between groups Ia and II MS afferent identities only emerge during postnatal development. Interestingly, in previous studies we demonstrated that a disruption of normal muscle spindle development results in the failure to induce or maintain expression of Heg1[28], a molecule we now find to be restricted to proprioceptors that innervate muscle spindles. Nxph1 was unaffected in animals that lack spindles[28], indicating that spindle-derived signals may inform some but not all aspects of MS afferent identity. This suggests that other qualitative or quantitative differences in mesenchymal, muscle, or tendon signals, and/or differences in activity levels may determine the full complement of muscle spindle afferent phenotypes.

Transcriptional differences between MS and GTO afferents emerge at an earlier developmental stage and can be observed as early as p0. At this stage, expression of the transcription factor Pou4f3 is selectively maintained in prospective group Ib GTO afferents. However, we find no evidence that $Pou4f3^+$ neurons derive from a specific subset of e14.5 progenitors, indicating that the Pou4f3-group Ib lineage relationship first appears during the e14.5/p0 interval after the afferents reach their nascent receptor targets. The absence of a clear lineage relationship between mature and early embryonic proprioceptors is observed for a broad panel of pSN subtype-selective molecules and parallels a recent study indicating that the transcriptomes that define adult somatosensory neuron subclasses bear little to no resemblance to the transcriptomes of early embryonic (e11.5) somatosensory neurons[26]. The notion that proprioceptor subtypes may emerge relatively late in development reinforces the idea that MS or GTO afferent subtype selection is influenced by extrinsic signals[26,28].

The initial broad and overlapping expression patterns of many transcription factors within embryonic somatic sensory neurons, and their apparent regulation by extrinsic signals, may confer additional flexibly to respond to subtle changes in peripheral target tissues.; differences in transcription factor expression levels can be rapidly amplified through a multitude of transcriptional targets. The role of peripheral signals in the regulation of neuronal identity may be a general feature of sensory systems given that they interface between the external environment and the brain. Within the context of the proprioceptive sensory system such a mechanism may facilitate adaptations to specific functional demands on the motor system. As such, a better understanding of the molecular mechanism by which Pou4f3 expression is selectively maintained in GTO afferents but not MS afferents should provide general insights into the developmental logic of sensory neuron subtype selection.

Taken altogether, these findings provide important insights into the organization, molecular basis, and development of MS and GTO afferent proprioceptor subtype identities, and should form an important foundation for genetic studies aimed at further understanding the role of these afferents in coordinated motor control.

## Methods

**Animal husbandry and mouse strains.** Runx3:FlpO animals were generated through homologous recombination of a Runx3:FlpO targeting vector in MM13 ES cells. Successful recombinants were identified through Southern blot analysis. Rx3: FlpO transgenic animals (heterozygous and homozygous) were identified through genotyping PCR analysis. Primers amplifying the FlpO allele are FlpO-For: 5′-GCA TCTGGGAGATCACCGAG-3′ and FlpO-Rev: 5′- GCCGTTCCAGGCGGGGTAT CTG-3′, which result in an 850 bp product. Primers used to distinguish homozygous from heterozygous or wild type animals: Rx3ex6 (F): 5′-GCGCCCTACCAC CTCTT-3′ and Rx3ex6(R2): 5′-TGGGAGCCACTGCCAGCTCTG-3′. These primers result in a 400 pb product in heterozygous and wild-type animals, but no product in homozygous mutant animals. Other mouse strains used were PV:Cre[30], Ai65D[32], RCE:FRT[31], Ai14[42], Mapt:eGFP-nLZ[30], TrkC:tdTomato[48], Calb2-IRES-Cre[38], Tac1-IRES2-Cre-D[41], and Pou4f3KOAP[39]. Animals of both sexes were used for all experiments. Age of animals at time of analysis is indicated per experiment. Animals were given ad lib access to water and food. All experiments were performed according to National Institutes of Health guidelines and approved by the Institutional Animal Care and Use Committee of Columbia University.

**Fluorescence activated cell sorting (FACS) of dissociated pSNs.** DRG from adult ($p > 56$), adolescent (p12), neonatal (p0), and embryonic mice (e14.5) of either sex were dissected (1 h maximum) in ice-cold Hank's balanced salt solution (HBSS) and collected, on ice, in HBSS supplemented with 0.75% horse serum (HS). The total number of animals (of either sex) used for experiments was six for adult samples, four for p12, six for p0, and two for e14.5. Following dissection, DRGs were centrifuged at low speed and washed once with ice cold HBSS prior to dissociation though enzymatic digestion using Papain followed by Collagenase/Dispase[35,59]. Papain digestion step consisted of 3 ml HBSS with ~16 units/ml Papain (Worthington), 0.83 mM L-Cysteine, 0.42 mM EDTA, and 20 units DNAse I/ml (Roche). Collagenase/Dispase digestion was with 3 ml HBSS containing 1,066 units/ml Collagenase IV (Worthington), 4 units/ml Dispase (Worthington), and 20 units/ml DNase I (Roche). Duration of digestion incubation times was 16 min for adult, 12 min for p12, 10 min for p0 and e14.5 (for both enzyme digestion steps). Solutions were exchanged by a low speed centrifuge step (4′ at 100 rcf in table top Eppendorf centrifuge) and aspiration. After Collagenase/Dispase digestion, enzyme solution was replaced with 500 μl HBSS supplemented with 20% HS and 20 units DNAse I, and DRG/cell suspension was dissociated by slow mechanical trituration using a 200 ul pipetman (~50 times). Following dissociation, cell suspension was incubated with 2 μM Calcein Blue (Cell-Trace™ Calcein Blue, AM; Invitrogen) for 15 min at room temperature. After Calcein Blue labeling, cells were spun down (4′ at 100 rcf) and resuspended in 500 μl sorting solution (HBBS, 1% HS, 20 units/ml DNASE I) supplemented with 0.01 μM Sytox red (SYTOX™ Red Dead Cell Stain; Invitrogen). Just prior to FACS, dissociated cells were passed through 40–70 μm gauze filters to clear remaining cellular aggregates. FACS of fluorescently-labeled (tdTomato+) neurons was performed at 12 psi, using a Becton Dickinson FACSAria (SORP model, 5-laser, 20 parameter), equipped with a 130 μm (p0, p12, p56) or 100 μm (e14.5) nozzle, and using 586/15 (tdTomato), 450/50 (Calcein Blue), and 670/30 (Sytox red) filter sets and a gating strategy as described in Supplementary Fig. 15. Fluorescent neurons were directly deposited in 96-well LoBind plates (Eppendorf) prefilled with 7.5 μl lysis buffer/well. Lysis buffer contained 0.2% Triton X-100 (Sigma), SUPERaseIN (1 U/μl) (ThermoFisher), 2 mM deoxyribonucleotides (dNTPs) (ThermoFisher), and 2 μM reverse transcriptase (RT) primer

(Integrated DNA Technologies)[60]. Plates were stored at −80 °C until processed for cDNA generation, library preparation and RNA sequencing.

**Single cell cDNA generation, library preparation, and RNA sequencing**. Single cell cDNA and library preparation was performed by the Columbia JP Sulzberger genome center as described previously[60]. In brief, after primer annealing (72 °C for 3 min), reverse transcription (RT) was performed by adding 7.5 µl of RT mix to each well. RT mix contained 2 M betaine (Affymetrix), 2× Protoscript Buffer (New England Biolabs), 12 mM MgCl₂ (ThermoFisher), 10 mM dithiothreitol (ThermoFisher), 5.3 U of Protoscript II Reverse Transcriptase (New England Biolabs), 0.53 U of SUPERaseIN (ThermoFisher), and 2 µM Template Switching Oligo (Integrated DNA Technologies). Reverse transcription was performed at 42 °C for 90 min, followed by 10 cycles of 50 °C for 2 min, 42 °C for 2 min, 70 °C for 10 min, followed by a 4 °C hold. Excess primers were removed by adding 2 µl of Exonuclease I (ThermoFisher) mix to each well (1.875 U of ExoI in water) and incubating at 37 °C for 30 min, 85 °C for 15 min, 75 °C for 30 s, and 4 °C hold. Following RT reactions, all wells were pooled into a single 15-ml falcon tube, and complementary DNA (cDNA) was purified and concentrated using Dynabeads MyOne Silane beads (ThermoFisher) according to the manufacturer's instructions. The cDNA was split into duplicate reactions containing 25 µl of cDNA, 25 µl of 2× HIFI HotStart Ready Mix (Kapa Biosystems), and 0.2 M SMART PCR Primer and polymerase chain reaction (PCR) mix. cDNA was amplified as above, and duplicate reactions were combined and purified using 0.7 volume of AMPure XP beads (Beckman Coulter). The amplified cDNA was visualized on an Agilent TapeStation and quantified using a Qubit II fluorometer (ThermoFisher).

Sequencing libraries were constructed using Nextera XT (Illumina) as described in ref. [60]. A custom i5 primer was used (Nextera PCR) with 0.6 ng of input cDNA, and 10 cycles of amplification were performed. Unique i7 indexes were used for each plate. After amplification, the library was purified with two rounds of AMPure XP beads, visualized on the TapeStation, and quantified using the Qubit II fluorometer. Libraries were sequenced on an Illumina NextSeq 500 using the 75-cycle High Output Kit [read lengths 26(R1) × 8(i) × 58(R2)]. Custom sequencing primers were used for Read 1 (SMART_R1seq and ILMN_R1seq; see ref. [60]). With each plate, we targeted ~70 million reads. Library pools were loaded at 1.8 pM with 30% PhiX (Illumina). All sequencing data is available through the NCBI GEO database (accession # GSE162263).

**Single cell transcriptome analysis**. Analysis of single cell RNA sequencing datasets was performed as described previously[61,62].

*Alignment of sequencing reads*. Raw sequence reads were aligned to the mm10 mouse reference genome with transcriptome annotation derived from NCBI. Alignment was performed using the STAR aligner, and reads were demultiplexed by Plate-seq barcode and collapsed by Unique Molecular Identifiers to obtain final transcript counts for each cell. For visualization of gene expression and calculation of gene-gene correlations, we normalized UMI counts to CPM (counts per million). For each cell, the number of UMIs for each gene is divided by the total UMI count for the cell and then multiplied by 10^6 to obtain CPM values. For differential gene expression among clusters we used the raw UMI count values, since edgeR performs its own distribution-specific normalization.

*Filtering of satellite cell contaminants*. To avoid clustering cells with significant oligodendrocyte and glial gene contamination, we set the following thresholds, based on the distributions of transcripts over the entire data set: Mbp Transcripts per million (CPM) < 10,000; Apoe Transcripts per million (CPM) < 20.

*Cluster analysis*. After alignment and initial filtering of cells for non-neuronal signatures, we performed a bootstrapped, iterative clustering approach, using the hicat R package as outlined in Tasic et al.[52]. This clustering algorithm proceeds as follows: (1) Select all cells. (2) Select top 6,000 genes with the highest variance/mean (Fano Factor) ratio. (3) Perform a modified Weighted Gene Network co-expression Analysis to identify modules of genes using a cutheight of 0.995 and soft-threshold power = 1. (4) When no gene modules with co-expression beyond chance are identified, terminate the iteration of clustering. Otherwise, cluster cells in the reduced dimensions corresponding to the eigengenes of the gene modules, using hierarchical clustering with Ward's Method and a distance metric defined by cell-cell correlation in eigengene space. (5) Test robustness of clusters by summing the negative log10 p value of differentially expressed genes (defined by the R limma package) among each pair of clusters. (6) Merge clusters for pairs of clusters with a sum of negative log10 p values < 100. We repeated steps 2–5 for each additional cluster identified until no further subclusters were identified. We next repeated steps 1–6 100 times using randomly selected subsets comprising 80% of all cells to generate a co-clustering matrix, where each entry represents the proportion of times a given pair of cells was found in the same cluster across 100 iterations. For each pair of clusters, we determined the mean co-clustering value over all cells belonging to both of the clusters. Clusters with mean co-clustering values >0.25 were merged.

*Differential gene expression and marker identification*. We used edgeR on the UMI count values to perform differential gene expression and marker identification[63,64]. We calculated differential expression across all pairs of clusters. A subset of combinatorial marker genes was selected for display (in Fig. 2) based on identification across multiple pairwise comparisons.

*t-SNE visualization*. Cells were visualized in t-SNE space using all differentially expressed genes across clusters (FDR-adjusted p value <0.05). After subsetting on these genes, we calculated the top 20 principal components on the log10(CPM + 1) transformed data. These PC values were input into the Rtsne package to calculate t-SNE coordinates for visualization. Cells were not clustered in t-SNE space.

*GO analysis*. Go analysis was performed for differentially expressed genes between each pair of clusters from the P56 cells. For each pair of clusters, we use edgeR to find differentially expressed genes, up in cluster 1 and down in cluster 2. The opposite comparison (down in cluster 1, up in cluster 2) was run as a separate gene list. Differentially expressed genes were filtered to those with log-Fold Change >1. The differential gene list was run through topGO with the following parameters: nodeSize = 5, annot = annFUN.org. Output of GO categories was filtered using the following criteria: (1) At least five overlapping genes between the differential gene list and the GO category, (2) Fisher.elim and Fisher.classic adjusted p values < 0.01, and (3) Category is related to neurons and neuronal processes (manual filtering). The final output consisted of a list of significant GO categories for each cluster (genes upregulated relative to each other cluster, in pairwise fashion).

*Longitudinal expression analysis*. For the cells obtained from the e14, p0, and p12 time points, the clustering was carried out as with the p56 data, and marker genes identified in the same way. For the gene-correlation analysis, we selected a subset of differential genes from p56 and calculated the Pearson correlation coefficient of the log(CPM + 1) value for each pair of genes from this gene subset over all the cells from a given time point. The plots in Fig. 7 reflect these correlation values, arranged according to the hierarchical clustering of these genes (average linkage) in the p56 data. A web-based searchable database is available at https://vmenon.shinyapps.io/proprioceptors_scrnaseq.

**Immunohistochemistry**. Immunohistochemistry was performed on cryostat sections (15–30 µm) or whole-mount muscle tissue as described previously[27]. For p8 and older tissues, animals were anesthetized using Avertin (1.25%) and perfused by transcardial perfusion with PBS (~20 ml, adjusted based on age of animal) and 4% PFA (in PBS; ~20 ml). Tissues were dissected in PBS and post-fixed for 2 h (<p12 and muscle tissue) to o/n (p12 and older tissues) in 4% PFA in PBS at 4 °C. (Please note: for staining with Pou4f3 antibody, tissues were perfused with 2% PFA in PBS and post-fixed for 30 min at RT.) Following fixation, tissues were washed in PBS (3 exchanges of 10 min each) and, for cryostat sections, equilibrated in 30% sucrose (in 0.1 M PO4 buffer), embedded in Tissue-Tek O.C.T. (Sakura Finetek), and stored at −80 °C until use. Sections were cut on a Leica CM3050S cryostat and collected on Superfrost plus slides (Fisherbrand) and stored air tight at −80 °C. When proceeding directly with immunostaining, slides/sections were air dried for ~15 min and rinsed once with PBS (5′) before o/n incubation (at 4 °C) with primary antibody in PBS, 1% BSA, 0.1% Triton X-100. Following primary antibody incubation, slides were washed twice in PBS (6–8 min/wash) and subsequently incubated with secondary antibody for 2 h at RT. Following incubation with secondary abs, slides were washed with two exchanges of PBS (6–8 min each) and cover slipped with fluoromount-G (Southern Biotech) or Vectashield (Vector labs). For whole mount muscle tissue, following PFA fixation and wash with PBS (in 24-well plates), muscles were dissected free (and teased into smaller segments depending on muscle size) and incubated in ice cold methanol (100%) for 5–10 min. After methanol permeation, muscle tissues were washed in PBS (4 exchanges of 5, 10, 15, and 15 min; at RT) before incubation in blocking solution (PBS, 1% BSA, 0.3% Triton X-100) for at least one hour under gentile agitation at RT or o/n at 4 °C. Incubation of primary antibodies was performed in 24 well plates (250 µl antibody soln./well) for 2–3 days at 4 °C under gentile agitation. After primaries, muscles were washed at RT for 3 × 5 min and 5 × 1 h in blocking solution and subsequently incubated with secondary antibody (for 1–2 days at 4 °C). After staining with secondary antibodies, muscle were washed as before with the final two washes performed in PBS without BSA/Triton X-100. Muscles were kept in PBS at 4 °C until mounted in Vectashied (Vector labs). Primary antibodies used in immunohistochemistry were: Rb anti-Calbindin D-28k (1:2,000) (Swant), Rb anti-Calretinin (Calb2; 1:2000) (Swant), Gp anti-tdTomato (1:32,000)[28], Rb anti-dsRED (1:1000) (Clontech), Rb anti-GFP (1:1000) (ThermoFisher), Shp anti-GFP (1:500) (AbD Serotec), Rt anti-Troma1 (1:100) (DSHB), Rt anti-Substance P (1:200) (Santa Cruz Biotechnology), Rb anti-Rx3 (1:50,000)[65], Gp anti-Rx3 (1:16,000)[66], Ck anti-PV (1:30,000)[27], Gp anti-Islet 1 (1:20,000)[67], Rb anti-vGlut1 (1:16,000)[68], Rb anti-VGlut2 (1:500) (Synaptic Systems), Rb S-100 (1:400) (Agilent), Ck anti-Brn3c (Pou4f3) (1:50)[69], Ck anti-b-galactosidase (1:5,000) (Abcam), Rb anti-PCDH8 (1:250) (Millipore). Fluorophore-conjugated secondary antibodies generated in donkey (Jackson Immuno Research Laboratories) were used at 1:1000 (FITC) or 1:500 (Cy3, Cy5) in PBS, 1% BSA, 0.1% Triton X-100. Images were acquired on LSM510 or LSM700 confocal microscopes (Carl Zeiss).

**Alkaline phosphatase labeling**. Alkaline phosphatase staining was performed as described previously[70]. In brief, tissues were fixed as described above and incubated at 68 °C for 90 min to inactivate endogenous alkaline phosphatase activity. Subsequently, tissues were washed twice (10 min each in AP prestaining buffer (0.1 M Tris, 0.1 M NaCl, 50 mM MgCl2, pH 9.5) prior to incubation with AP staining buffer (prestaining buffer supplemented with 0.34 g/ml nitroblue tetrazolium (NBT) and 0.175 g/ml 5-bromo-4-chloro-3-indolyl-phosphate (BCIP) (Invitrogen). Alkaline phosphatase staining reaction proceeded o/n at RT. After staining, tissues were washed three times for 20 min in PBS, 0.1% Tween 20 and postfixed in PBS with 4% paraformaldehyde o/n. After washes with PBS, samples were dehydrated through an ethanol series and then cleared with 2:1 benzyl benzoate (BB)/benzyl alcohol (BA). Images were acquired using a Zeiss Axioscope 2.

**RNAscope**. RNAscope analysis was performed using the ACDbio RNAscope kit for nonfixed tissue (ACD #320851) based on ref. [71]. In short, DRG extracted from adult mice were fixed for 15 min in 4% PFA, washed in PBS, and allowed to equilibrate in 30% sucrose for 2 h before embedding in OCT (Tissue Tek; Sakura Finetek). DRG were stored at 80 °C until use. On day of experiment, DRG were sectioned at 20 μm, dried for 1 h at 30 °C, washed in PBS, dehydrated in a series of 50%, 70%, and 100% EtOH steps, and pretreated with protease IV digestion for 30 min (at RT). Hybridization was performed in a humidified oven (ACDbio) at 40 °C for 2 h. Following hybridization, tissues were washed and processed for probe amplification and detection using Amp1 (30 min), Amp2 (15 min), Amp3 (30 min), and Alt4 (A, B, or C; 15 min). All wash steps were performed at 40 °C in the same humidified hybridization oven. Probes used in experiments were: Mn-Agpat4 (cat# 489901-C2), Mn-Cdh13 (cat# 443251-C3), Mn-Chad (cat# 484881-C1), Mn-Colq (cat# 296211-C2), Mn-Fam196a (cat# 505471-C2), Mn-Gabra1 (cat# 435351-C1), Mn-Gabra3 (cat# 435021-C3), Mn-Gabra5 (cat# 319481-C1), Mn-Grin3a (cat# 551371-C1), Mn-Heg1 (cat# 510581-C2), Mn-Hpse(cat# 412251-C1), Mn-Npy1r (cat# 427021-C1), Mn-Nxph1 (cat# 463401-C1), Mn-Pcdh8 (cat# 558101-C2), Mn-Pcdh17 (cat# 489901-C2), and Mn-Pvalb (cat# 421931-C3) (all Advanced Cell Diagnostics, INC). When combined with immunostaining, following Alt4 incubation step, slides were rinsed with PBS once and processed as described above (*Immunohistochemistry*). Images were acquired on a Zeiss LSM700 confocal microscope.

**Electrophysiology**. Electrophysiological experiments were performed at ~37 °C on dissociated DRG cultures. Dissociated neurons were obtained from adult mice (age range 6–9 weeks old) and cultured 24 h prior to the recording session. In most experiments, dissociated neurons were prepared as described above (*Fluorescence activated cell sorting (FACS) of dissociated pSNs*) and cultured o/n on Poly-Ornithine/Laminin-coated glass coverslips in DMEM/F12 supplemented with 10% horse serum and 1x penicillin/streptomycin. In few experiments, mice were perfused with oxygenated ice-cold ACSF/choline solution as in ref. [72], supplemented with Tetrodotoxin (TTX; 1μM), prior to dissection of ganglia in oxygenated dissection solution, including Kynurenic acid and TTX, as described[72]. In these experiments, enzymatic digestions were extended to 30 min each for Papain and Collagenase/Dispase (both performed under constant oxygenation), and dissociated neurons were cultured in MEM, 10% horse serum, 10 mM HEPES, 0.45% Glucose, 1x penicillin/streptomycin. Post-hoc analysis showed no difference in neuronal recordings for the neurons obtained from the different preparations and when appropriate data were combined.

Whole cell patch clamp current recordings were performed using a Multiclamp 700B amplifier (Molecular Devices, Sunnyvale, CA, USA). Signals were acquired at frequency of 100 KHz and filtered at 10 KHz. Borosilicate electrodes were pulled using a puller P-1000 (Sutter instrument) with resistance ranging between 4 to 8 MΩ and filled with an intracellular solution containing (in mM): 130 K-gluconate, 10 NaCl, 10 HEPES, 1 EGTA, 1 MgCl2, 0.1 CaCl2, and 1 Na-GTP. pH was adjusted to 7.2 with KOH. Osmolarity was adjusted to 290 – 295 mOsmol Kg$^{-1}$ with sucrose. The extracellular solution contained (in mM): 145 NaCl, 3 KCl, 1.5 CaCl2, 1 MgCl2, 10 D-glucose and 10 HEPES. pH was adjusted to 7.4 with NaOH. Osmolarity ranged between 310 – 315 mOsmol Kg$^{-1}$. The junction potential was corrected and taken into account for subsequent analysis. Only tdTomato positive cells were recorded, identified by epifluorescence using a Leica (DM 6000FS) upright fix-stage microscope. The diameter of the cells was measured with the Leica Application Suite X software (Leica Microsystems). After whole cell access was established, seal parameters were recorded and only cells with input resistance ($R_{in}$) over 75 MΩ were analyzed, based on the value for $R_{in}$ reported in other reports for large DRG neurons and specifically for PV$^+$ neurons[44,73]. The input resistance for each cell was obtained from the slope of a steady-state (linear) current-voltage plot in response to a series of negative current injections. After establishing the resting membrane potential in current clamp mode, cells were held at −60 mV and a series of current steps of 50 pA (duration of 250 ms) was injected to induce action potentials (APs), in order to reveal the firing pattern (assigned as: single AP, burst or phasic). Signals were acquired using a Digidata 1440 A (Molecular Devices, Sunnyvale, CA, USA) controller by pClamp 10.3 v software. To ensure supramaximal stimulation, the current injection was increased to 200 pA over the rheobase. The parameters measured were: (1) capacitance, (2) resting membrane potential ($V_m$), (3) rheobase, (4) voltage threshold for AP induction, (5) AP amplitude, and (6) AP half-width (see Fig. 6 and Supplementary Fig. 11).

Additionally, we performed two protocols to identify the maximum firing frequency and to determine the ability of neurons to induce action potentials at different frequencies of stimulation following a small current step injection. For the first protocol, cells were stimulated at supra-threshold values using 200 to 400 pA over rheobase with a continuous current for 2.5 s (Fig. 6c). The number of APs per second was calculated. For the latter, a series of current injections with a short duration (5 ms) at different frequencies were performed using currents over 200 to 300 pA of rheobase for a total duration of 2.5 s in each series. The frequencies analyzed ranged from 4 Hz to 160 Hz. The total number of APs was quantified. Experiments using pharmacological blocking for Kv channels were performed stimulating the cells with the same current-clamp protocol of 50 pA steps as described above. After the identification of their firing pattern, neurons were exposed to 20 nM of DTXα (Alomone D-350) in extracellular solution for 3–5 min, and firing responses were recorded during and after washout. Analysis of rheobase, AP amplitude, AP half-width (see Supplementary Fig. 11), and firing frequencies versus current injection up to +300 pA rheobase, were done in order to compare DTXα effects. All recordings were analyzed off-line with the software Clampex 10.1 (Molecular Devices Corp., Sunnyvale, CA, USA).

**Quantification and statistical analysis**. For whole DRG neuronal counts, analyses were performed on serial cryostat sections (30 μm) of individual ganglia, with each section counted. Counted cell bodies and nuclei only included those with near full-size diameter to avoid double counting of cells or nuclei at the edge of sections. Except where stated otherwise, a minimum of three DRG, obtained from three experimental or WT animals, were counted per segmental level and per experimental condition. Average counts/DRG and SEM were calculated through Sigmaplot (Systat Software). For neuronal counts on tissue sections obtained at rostral or caudal lumbar DRG (e.g. RNAscope, immunohistological analysis), individual sections (20 μm) from cervical, thoracic, or lumbar DRG were analyzed. The average number of neurons/section was calculated by dividing the sum of all counted cells by the total number of counted sections. To estimate the proportion of pSNs expressing a given transcript, marker expressing neurons were counted across multiple tissue sections (at least 6 per probe combination) and gated to PVRx3 (tdT$^+$) neurons. Percentage of PVRx3:tdT$^+$ neurons expressing a certain marker combination was calculated by dividing the number of marker neurons for each section with the number of PV$^+$ neurons for the same section. Average percentages and SEM were calculated through Sigmaplot (Systat Software). Statistical analysis, as assessed by Student's *t* test or Mann–Whitney *U* test, on all counted neuronal populations or cell diameters was performed using Sigmaplot (Systat Software). Significance was accepted for $p < 0.05$. Statistical analysis for electrophysiological recording experiments was performed using the software GraphPad Prism 6 (GraphPad, Inc.) and data are reported as mean ± standard error of the mean (S.E.M.) or as mean ± standard deviation (S.D.). *Kolmogorov-Smirnov* test was used to test normal distribution. Significance was set as *$p < 0.05$, **$p < 0.01$ and ***$p < 0.001$ as assessed by Paired *t*-test or One-way ANOVA; Tukey's post hoc test.

**Reporting summary**. Further information on research design is available in the Nature Research Reporting Summary linked to this article.

## Data availability
All sequencing data are available through the NCBI GEO database (accession number GSE162263). A web-based searchable database on proprioceptor subtype transcriptomes is also available at https://vmenon.shinyapps.io/proprioceptors_scrnaseq. Requests for (information on) resources and reagents should be directed to J.C.N. Source data are provided with this paper.

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

## Acknowledgements

We thank Barbara Han, Monica Mendelsohn, and Nataliya Zabello (Columbia Zukerman Mind and Brain Behavior Institute) for help with generating the *Runx3:FlpO* mice, David Ginty (Harvard) for providing the *TrkC:tdTomato* mice, Mike Kissner (Columbia Stem Cell Institute Flow core) for help with FACS plate sorting, Erin Bush (Columbia JP Sulzberger genome center) for help with single cell RNA sequencing, and Siddarth Bejugama for help with initial RNA in situ analyses. We also greatly appreciate the time and effort of Niccolo Zampieri, Peter Sims, Eiman Azim, and Marcela Carmona, for providing comments on the manuscript. This research used the Genomics and High Throughput Screening Shared Resource, funded in part through the NCI Cancer Center Support Grant (NIH P30CA013696). In addition, G.Z.M. was supported by NINDS (NIH R01NS078375), The NIH Blueprint for Neuroscience Research, NIAA, NINDS (NIH R01AA027079), The SMA Foundation, and Target-ALS; T.C.B. was supported by NEI through the Intramural Research Program; V.M. and J.C.N. were supported by NINDS (NIH R01NS106715 to J.C.N.), and by the Thompson Family Foundation Initiative (J.C.N.).

## Author contributions

J.C.N. conceived of the study, designed, and performed all experiments with help from K.M.O., and analyzed data. D.M.F.P. and G.Z.M. performed electrophysiological experiments and analyzed data. T.C.B. provided valuable unpublished reagents. V.M. analyzed single cell RNA sequencing data. J.C.N. wrote the manuscript with input from all other authors.

## Competing interests

The authors declare no competing interests.
