## [Peer Review File · Nature Communications]

Reviewer #1 (Remarks to the Author):

In this study, the authors perform single-cell sequencing of genetically labeled proprioceptors, identifying molecular markers for the different physiological classes (type Ia, Ib, II) and proposing that type II proprioceptors might be more heterogeneous than previously appreciated. They perform an analysis of the development of proprioceptor molecular diversity and determine that distinct proprioceptor gene expression patterns emerge relatively late in postnatal development. The primary strengths of this manuscript are that it is the first to characterize molecular profiles that correlate with different functional classes of proprioceptors. The genetically based deep sequencing of single neurons allows for the identification of molecular differences between proprioceptors that were not detectable in previously published analyses of sensory neurons. The authors identify markers and genetic tools that when used intersectionally might allow them to manipulate specific classes of proprioceptors individually, opening up many exciting new directions for the field.

However, there are some weaknesses that limit the potential impact of this work. First, while the unanticipated molecular diversity of presumed type II proprioceptors is heavily emphasized throughout the manuscript, there are no *in vivo* physiological or functional experiments that follow up on this finding from the sequence analysis. It is not clearly demonstrated that clusters 2 and 3 are type II proprioceptors. The evidence supporting this hypothesis is indirect (cell body diameters of *Nxph1+* neurons; relative proportions of different markers). The authors convincingly use *Calb2*, *Tac1*, and *Pou4f3* as markers to interrogate the functional identity of clusters 1, 4, and 5 respectively, and demonstrate that each largely labels a distinct class of proprioceptor endings. It is understandable that readily available reagents may not exist for the genes specific for clusters 2 and 3, but in the absence of data validating the identity of these clusters their significance remains ambiguous. Any insight into anatomical or functional differences between the three putative Type II clusters would go a long way to increase the impact of this study. For instance, the authors speculate that the different type II clusters may correspond to differences among afferents relaying information from axial or limb muscles. A more comprehensive analysis of marker genes *in situ* done at all segmental levels could support or rule out this hypothesis.

Related to this, the authors propose that the two functional classes of proprioceptors detected in their *in vitro* physiology experiments correspond to group I versus group II afferents, but this is not tested. *In vivo* physiological measures would be most impactful in addressing this possibility as well as addressing the issue about the presumed type II proprioceptor diversity, mentioned above. In light of the genetic tools that this study describes, one could try to specifically label Group Ia afferents using *Calb2-Cre*; *Runx3-FlpO* and determine if proprioceptors labeled this way display the rapidly adapting responses observed in some of their cells when all proprioceptors are labeled. Similarly, *Tac1-Cre* could be used intersectionally with *Runx3-FlpO* to label group II afferents and determine whether or not they show a tonic firing pattern, as predicted.

Specific comments:

1. Although the work presented is generally of high quality, some of the findings, including the extent of overlap between identified marker genes *in situ* is described in qualitative terms ("generally co-expressed," "often overlaps with") and supported by representative images (Fig. 3, Supplemental Figure S5). Quantification is limited to the % total *tdT+* neurons positive for each marker gene. Adding the quantification of the % overlap between putative subset-specific markers would greatly improve this analysis.
2. In a similar fashion, more quantification to demonstrate the efficiency and specificity of newly identified genetic tools would be helpful in determining their usefulness for various types of experiments. For example, the authors show that *Calb2-Cre*; *LSL-GFP* labels many but not all muscle spindles: what % of spindles are labeled? Similarly, *Tac1-Cre* is shown to label afferents at the polar ends of muscle spindles (type II) but it is noted that it can also label afferents at the equatorial regions of spindles (type Ia): at what frequency are these two classes observed? What proportion of all type II afferents does *Tac1-Cre* efficiently label?
3. It may be beyond the scope of the current study to characterize the contributions of different ion channels to the firing properties of proprioceptors observed *in vitro*, but one way to test the

author's hypothesis that the different potassium channels expressed in type I vs type II proprioceptors contribute to different intrinsic firing properties would be to use pharmacology to test the contribution of some of the candidate channels identified (for instance Kv1.1 and Kv1.2, which are more highly expressed in clusters 1 and 5, putative type I proprioceptors, and may contribute to their rapid adaptation to current injection). This would improve this section of the study, which otherwise remains descriptive and speculative.

4. The sequence analysis for the adult population relies on 208 neurons, which is not a large number for this type of work. The developmental analysis relies on even fewer neurons than the adult analysis (in particular at P0, 55 neurons) and it is therefore difficult to be confident that the determination of fewer clusters truly reflects decreased molecular diversity at this age and is not simply due to lower resolution in cluster identification.

5. The authors assume that early Pou4f3 (Brn3c) expression predicts future type Ib identity, but do not test this. A lineage tracing experiment would strengthen the resulting arguments and interpretation of the developmental sequencing data.

6. Is Supplementary Fig. 2g missing?

7. The abstract states: "definitive molecular correlates of these three muscle afferent subtypes remain unknown" and the Introduction states "definitive molecular signature for functionally distinct proprioceptive muscle afferents remains to be established" While this is true, defining molecular diversity as an endpoint may not be of sufficient impact for a research article, especially without definitive proof of their relationship to neuronal populations with defined physiological, functional or morphological properties, although it may be for a resource paper if it provides clear genetic tools for these populations and new molecular insights.

Reviewer #2 (Remarks to the Author):

This manuscript reports the first evidence of unique molecular signatures for different classes of muscle proprioceptors. These specialized mechanoreceptors are indispensable to kinesthetic sense as well as to correction of and prediction about sensory-guided movement, yet the specific roles of different classes of muscle proprioceptors in these processes remain incompletely understood, as do cellular mechanisms involved in their sensory encoding and network operations. Findings reported here introduce a new experimental tool/strategy having real potential to substantively improve understanding in these fields. The key result is demonstration that unique clusters of molecules predict the identities of proprioceptor sub-classes independently validated by morphological features of the sensory receptors they innervate. These determinations were made with rigorous and creative applications of genetic strategies coupled with single-cell RNA sequencing and bioinformatics approaches that uncovered molecular/histological means for labeling and distinguishing proprioceptor subtypes. In a particularly significant observation arising out of the first application of this approach, lineage analysis at different developmental stages leads the authors' to suggest that the unique transcriptional profiles of the proprioceptor subtypes are induced by contact with their receptor targets. Experimental design and rigor are carefully detailed, the primary and supplemental data and analyses are sound, incisive and convincing, the findings are clearly presented, properly supported by relevant literature, and carefully interpreted in general. A few comments are offered below suggesting reasons for tempering some conclusions and providing more detailed presentation of some data. These are relatively moderate flaws that do not diminish the likelihood that delineation of muscle proprioceptors reported here will prove extremely valuable to studies aimed at determining their roles in control of sensory-guided movement and more specifically, their contributions to neural networks, and specificity in regeneration.

Major comments

- My major comment has to do with statements and conclusions claiming the data show that neurons with different molecular expression correspond to or were tested for being functionally distinct subtypes (see Abstract, Intro summary page 5, Subtitle page 9, text pages 11, 12). These

statements over-reach the data. Functional responses to mechanical stimulation were not measured but only inferred from the anatomy of receptor innervation and terminal endings. The 3 molecular clusters assigned to group II may or may not be functionally distinct, and all muscle proprioceptors – Ia, II, spindle unclassified and Ib – exhibit some functional redundancy. Comments and conclusions related to assigning molecular signature to afferent function require more precision and nuance.

- The whole-cell electrophysiological recordings of membrane and firing properties have value in describing the range physiological properties of PVRx3;tdT neurons. However, there are no data demonstrating how slowly vs rapidly adapting neurons were associated with neuronal subclass or with different potassium channels, which might have been examined from voltage-clamp study K currents. For these reasons, speculation in Results (top paragraph p. 14) should be limited and restricted to Discussion.
- Data demonstrating that a neuron's expression of unique molecules predicts the receptor or receptor region it innervates is the central finding. It is important then to provide data on the number of neurons and their receptor connections examined per cluster and the number of neurons that connected with the 'wrong' receptor. These interests may even warrant a table.
- It is arguable that the authors assign too much significance to the diversity in MS subtypes. Diversity in molecular identification fits well with the continuum observed in the spike train encoding properties of Ia, II, and unclassified MS afferents. The appearance of 3 group II clusters may reflect the large range in mechanosensory encoding by group II, assuming that the 3 group II clusters are actually distinguishable physiologically. The observation have more to do with imprecision/arbitrariness in physiological classification than is does with meaningful differences.

Minor comments

Page 6: "... A few afferents that appear to project to deep dorsal horn targets, ...". This statement needs revision recognizing that group Ia, II and Ib all send dense synaptic projections into deep dorsal horn (LV, VI) in cat (Brown and Fyffe, 1978) rat (Vincent et al. 2017), and mouse (Levine et al. 2014).

Timothy Cope

Reviewer #3 (Remarks to the Author):

The manuscript by Katherine Oliver and colleagues presents the first thorough, complete and detailed single-cell transcriptomic analysis of proprioceptive afferents in the adult mouse and also during embryonic and early postnatal development. Proprioceptor afferents are critical for movement and the sense of self and space, and although they have been thoroughly studied physiologically, their molecular/genetic identify has remained largely unknown. This is in part because, as described in this study, their unique properties arise not due to specific genetic lineages and early fating, but are regulated by extrinsic factors during postnatal development. Thus, they do not have a defined specific genetic code that could be used for their identification throughout development. This has been suspected for some time now in the field, but this study finally makes the full and complete case for this fact. The work now defines their genetic differences in the adult after developing their specific properties and constitutes a tour-de-force with impressive validation of many of the genes identified. It is full of new findings that are the culmination of this very long effort by several labs seeking to uncover the molecular identity of these afferents. Where others failed the lab of Dr. Joriene de Nooij succeeded, and in a very impressive manner.

The main new findings to highlight area that: 1) proprioceptors that are usually classified in three physiological types (Ia, Ib, II) can in fact be divided in 5 molecular/genetic classes considering three subclasses of type II afferents adding diversity to these important and seldom studied afferent type; 2) revealing for the first time clear molecular markers to distinguish Ia and Ib afferents, with some being quite surprising (VGLUT2 in Ia afferents); 3) finding that this genetic/molecular diversity arises during maturation of these afferents, i.e., different

proprioceptive afferents are not derived from specific lineages but arise by developmental factors influencing their maturation after they arrive to muscle (this confirms a previous report from this same group suggesting that key proprioceptive genes are dependent on muscle factors); 4) the study uncovers a diversity of K channel expression patterns that may underlie the physiological differences among these afferents; 5) finally, an intriguing afferent-type specific diversity in GABAA receptor subunit expression was also found suggesting that mechanisms that control the transfer of sensory information from each of these afferents to the CNS could be controlled by presynaptic mechanisms that differ in pharmacology and kinetics and could be dissected out in future studies. These results are all extremely significant and will be highly used and likely cited by future investigators. They also provide entry points for developing new animal models to specifically dissect the individual functional significance of each of these afferents. This has not yet been possible. If the results and conclusions from this study are confirmed and further validated by others, it is my view that this study will constitute one of the major advances in the study of proprioception in the last decade. In any case this should be a valuable resource for future research and the whole data set will be made available as indicated in the manuscript (although the exact details are not yet defined) (Page 12: An interactive web application to search for individual genes in proprioceptor subtypes will be made available at xxxx).

However, there are also significant issues that need to be fixed. Many are detailed in a point by point manner below. However, the major weakness of this study is the poor correlation between the physiological data and the molecular subtypes identified. It is unclear why a direct study revealing the molecular identity of the recorded cells was not done. The more indirect and speculative correlations proposed might be valid, but they make a much less compelling case for the conclusions proposed. Moreover, since the two major firing types described are not directly correlated with any of the differences in channel expression revealed by the genetic data.

In addition, there are several other significant issues throughout the manuscript that require the authors attention. These are detailed in the following points.

1. Counting method: Different figures show data for either number of DRGs or number of sections. It is unclear why there should be two different ways of sampling data. Why not analyze data always per DRG? Since sectioning increases sampling variability, this approach would diminish the large variability observed in many graphs. This variability sometimes goes from 0% to 100%, which it can be interpreted as the sections being highly variable in the number of cells they contain for analysis. Setting the data points per section seems not representative of the real distribution. Can all figures be set per DRG analyzed? Methods indicate that minimum of 3 DRG were sampled per experiments (with a few exceptions). These should be enough for statistical comparisons when enough sections per DRG are studied to obtain accurate DRG data points.

2. Page 6 (also figure S1f): Projections to dorsal horn laminae do not rule out that an axon could be from a proprioceptive Ia afferent. Distal flexors project to dorsal horn laminae in the trajectory shown in the figure (see Figure 9 in: Ishizuka N, Mannen H, Hongo T, Sasaki S. 1979 J Comp Neurol.186(2):189-211. doi:10.1002/cne.901860206)

3. Page 6: "Intriguingly, we only observed tdT+ Merkel cell afferents in forelimb skin and not in the skin of the back."
What about hind limbs which are the focus of the paper?

4. Page 7: "PVRx3:tdT reporter provides an efficient and nearly exclusive means to manipulate or isolate proprioceptive muscle afferents."
Can this be supported with quantitative data? What percentage of PVRx3:tdT DRG cells are not proprioceptors?

5. Page 8: "Expression of Cdh13 and Crtac1 is biased to neonate proprioceptors that innervate dorsal (flexor) or ventral (extensor) limb muscle targets."
The correspondence between flexor and extensor actions and ventral vs dorsal limb muscle is full of exceptions and contradictions and should be abandoned. The gastrocnemius is ventral but is a flexor of the knee and an extensor of the ankle; the biceps is a flexor at the knee but ventral, the rectus femoris is a knee extensor also a hip flexor and dorsal. The vastus medialis is a pure knee

extensor but dorsal ... and so on. The developmental significant issue for axon trajectories and wiring is the location of developing the muscles, not whether they function as flexors/extensors across different joints. I understand this generalized equivalences became standard in the field, but that they keep being maintained is very perplexing.

6. Page 8: this patterns (not plural).

7. Page 10: "The pattern of CR+ Ia afferents in adult muscle raises the possibility that not all muscle spindles are innervated by a group Ia afferent."

This conclusion is quite a "stretch" since there is no other evidence provided for this possibility. Moreover, it is repeated in conclusions. The authors should be careful about this and provide additional supportive data since if this was true it would be a major finding. This should be easily tested by adding an alternative marker of Ia afferents in the spindle (VGLUT1 annulospiral endings around the central bag fibers). The experiment should be done before making such a dramatic conclusion. The alternative possibility of diversity in expression of CR by Ia afferents seems better supported by the single cell expression data in Figure 2g.

8. Page 11: "Consistent with previous observations that most DRG neurons express Pou4f3 at early embryonic stages, initially all pSNs (i.e. both MS and GTO afferents) are labelled by AP (Supplementary Fig. 7a-c)."

I do not think I follow the data presented in here. The data shown in Fig S7 is all at p2 and it seems that muscle spindles express little or any Pou4f3. Is there data at any other earlier time point that supports this statement of all pSNs expressing Pou4f3 before p2?

9. Page 14. "Based on previous observations that groups Ia and Ib afferents have dynamic (RA) response properties, we postulate that RA neurons correspond to groups Ia and Ib proprioceptors, while tonic neurons correspond to group II afferents."

I find this not very compelling; the dynamic and static responses to stretch and force of Ia, Ib and II afferents are encoded by ion channels at the terminal end in the spindle or GTO and not by ion channels at the cell body. As adequately discussed later in conclusions, Ia and Ib can evoke significant amounts of tonic firing to sustained stretch or tension. In muscle spindles the tonic firing of Ia afferents to sustained stretch can be as high or even higher than that of type II. The main differences are the dynamic responses. For a description of the the firing properties of these afferents in rodents consult the characterization of Vincent JA, Gabriel HM, Deardorff AS, et al. 2017J Neurophysiol.;118(5):2687-2701. doi:10.1152/jn.00497.2017. Considering this it seems necessary to obtain direct evidence that the neurons with single spikes have molecular features of Ia or Ib afferents and the one with tonic firing through the pulse represent type II.

That will be a very interesting finding and as discussed later parallels findings in Merkel afferents. However comparisons with Merkel afferents to justify Ia/Ib identity breakdowns in logic because Merkel afferents responses to skin indentation are slowly adapting and therefore comparison with Ia/Ib I(considered by the authors RA) vs II (considered by the authors as SA) is not straightforward. These problems could be solve by experimentation that directly reveals the identity of the recorded cells.

10. On VGLUT2 expression in Ia afferents: "However, the observation that adult group Ia MS afferents MS, in addition to vGlut1, also express vGlut2 was unexpected and may have important implications for studies that assess inputs to spinal neurons on the basis of the presumed differential expression of these two glutamate transporters between proprioceptors and spinal neurons."

I find this statement problematic in the sense that without qualification will create tremendous confusion. Quantification of VGLUT1 and VGLUT2 expression in the synaptic boutons of electrophysiologically identified and intracellular filled Ia afferents reported that both were present at significant levels above background, but while VGLUT1 was detected at >10 times background levels, VGLUT2 was barely above background (Alvarez FJ, Titus-Mitchell HE, Bullinger KL, Kraszpulski M, Nardelli P, Cope TC. J Neurophysiol. 2011;106(5):2450-2470. doi:10.1152/jn.01095.2010). Moreover, high sensitive EM ICC using postembedding colloidal gold in cryosubstituted specimens reported that many VGLUT1 synapses in the spinal cord also contain low levels of VGLUT2, but this was more difficult to detect using co-localization with immunofluorescence (Alvarez FJ, Villalba RM, Zerda R, Schneider SP. J Comp Neurol.

2004;472(3):257-280. doi:10.1002/cne.20012). Thus, VGLUT2-IR boutons detected with immunofluorescence can hardly be traced to Ia afferents, contrary to what is suggested here by the authors. VGLUT2 levels in the terminal might be too low. However, it might explain why neurotransmission is still preserved in the monosynaptic stretch reflex in VGLUT1 KOs. Mende M, Fletcher EV, Belluardo JL, et al. 2016Neuron.;90(6):1189-1202. doi:10.1016/j.neuron.2016.05.008.

11. Related to the above discussion, it is quite common for VGLUT1 systems to switch during development from VGLUT2 to VGLUT1. This has not yet been fully shown for proprioceptors. Does the data suggest that this is the case? Moreover was VGLUT1 detected? Is it possible that VGLUT2 low level expression in Ia afferents is a result of incomplete developmental downregulation?

Figures/Figure legends issues:

12. Figure 1. I am not clear about the necessity of including S100 immunolabelings. I think it is distracting from the main point.

13. Figure S1e. "while fragments of GFP+PV+ neurons (lacking a nucleus) can be detected (arrowheads), 'full size' (nucleated) GFP+PV+ neurons that lack RX3 are not observed." I am not clear what the authors are referring to in here and arrowheads are not shown.

14. Figure S1f,g I would remove the box graphs indicating averages and percentiles since these stats make no sense with $n = 2$. It says that $n = 2$ DRG, how many sections on average to estimate each DRG data point? Methods say that all sections were analyzed. Please clarify in figure legend.

15. Figure S1g. I do not see the need to show the percentage of PV+ and PV- Rx3+ neurons. It is a binary criterion that sums 100% therefore the data in both box plots is identical, just the inverse of each other. You only need to show one of them.

16. Figure S1i. Please refer to this figure in the appropriate section in the text. RPV- and RPV+ are confusing. Legend refers them as Rx3+PV+ and Rx3+PVoff. It will be best to be consistent in labeling between figure and figure legend.

17. Figure S5. The image panels are frequently mismatched with the descriptions in the results. Please review.

18. Figure 4j. Is the GTO labeled by VGLUT1(-ICC) as in figure or GFP+ GTO as in legend? If the second, how is that accomplished: VGLUT1-GFP?

19. Figure S4. In c, 173 cells are indicated as distributed in the five clusters. But adding all the cells in f, the number is 166. Why the discrepancy? Also throughout the manuscript it is never explained the depiction of the different clusters in figures like the f panel. What does it mean the overlapping regions between C1 and C2, C3 and C4 and C4 with C5?

20. Figure 2 vs S4. The % of cells in C1 is 31% in Figure S4 and 30% in Figure 2b. Are there small differences in the number of analyzed cells between these figures?

21. Figure 3i (ii) the legend indicates that expression of *Nxph1* and *Pcdh8* in this cell is comparable, but that is clearly not the case in the figure. That is not the case in Figure S4g, either. The conclusion that comparable expression of these two genes defines C2, C3, C4 seems not to be accurate in view of the data with RNAscope and the expression profile in Figure 2g.

22. Figure S6f,g. Both spindles display genetically labeled VR axons flanked by VGLUT1-IR endings in more polar regions. Was one of them supposed to show an example of a GFP+ axon not flanked by non-GFP+ VGLUT1-IR axons as in legend.

23. Figure S6i, What is the evidence that the genetically labeled CR axon belongs to a gamma motoneuron? Are there any end plates to show this axon is not sensory? If is indeed motor, could

it be a beta axon?

24. Figure 6c. Calbindin in motor axons. Calbindin is expressed by embryonic neurons and quickly downregulated during postnatal development but the time course of downregulation is pool dependent. In this sense the image in Figure 6c from a P12 might be more and exception than the norm. Please indicate so in the figure legend to avoid misleading the reader. Calbindin-ICC very infrequently labels NMJs. Moreover, the relevance of these data in this paper is not clear.

Please find enclosed the point-by-point rebuttal to our revised manuscript entitled “Molecular correlates of muscle spindle and Golgi tendon organ sensory afferents revealed by single proprioceptor transcriptome analysis”. We were pleased that all three reviewers were excited about our work and its future implications. We would also like to extend our gratitude to each of the reviewers for their thorough assessment and for providing helpful suggestions as to how to improve upon the earlier version of the manuscript. In the revised version, we have addressed the concerns raised by the reviewers by adding new results from additional experiments and expanding our data analysis. We also amended our conclusions when suggested by the reviewers. Below, please find each of the reviewers comments and our responses, as well as an explanation on how we have revised our manuscript to address the critiques. (Revisions in the manuscript are marked in blue font.) We are happy to answer any additional outstanding issues should there be a need.

Best wishes,
Joriene

REVIEWER COMMENTS

Reviewer #1 (Remarks to the Author):

In this study, the authors perform single-cell sequencing of genetically labeled proprioceptors, identifying molecular markers for the different physiological classes (type Ia, Ib, II) and proposing that type II proprioceptors might be more heterogeneous than previously appreciated. They perform an analysis of the development of proprioceptor molecular diversity and determine that distinct proprioceptor gene expression patterns emerge relatively late in postnatal development. The primary strengths of this manuscript are that it is the first to characterize molecular profiles that correlate with different functional classes of proprioceptors. The genetically based deep sequencing of single neurons allows for the identification of molecular differences between proprioceptors that were not detectable in previously published analyses of sensory neurons. The authors identify markers and genetic tools that when used intersectionally might allow them to manipulate specific classes of proprioceptors individually, opening up many exciting new directions for the field.

However, there are some weaknesses that limit the potential impact of this work. First, while the unanticipated molecular diversity of presumed type II proprioceptors is heavily emphasized throughout the manuscript, there are no *in vivo* physiological or functional experiments that follow up on this finding from the sequence analysis. It is not clearly demonstrated that clusters 2 and 3 are type II proprioceptors. The evidence supporting this hypothesis is indirect (cell body diameters of *Nxph1+* neurons; relative proportions of different markers). The authors convincingly use *Calb2*, *Tac1*, and *Pou4f3* as markers to interrogate the functional identity of clusters 1, 4, and 5 respectively, and demonstrate that each largely labels a distinct class of proprioceptor endings. It is understandable that readily available reagents may not exist for the genes specific for clusters 2 and 3, but in the absence of data validating the identity of these clusters their significance remains ambiguous. Any insight into anatomical or functional differences between the three putative Type II clusters would go a long way to increase the impact of this study. For instance, the authors speculate that the different type II clusters

may correspond to differences among afferents relaying information from axial or limb muscles. A more comprehensive analysis of marker genes by *in situ* done at all segmental levels could support or rule out this hypothesis.

We acknowledge that we inadvertently put too much emphasis on the molecular categories we believe constitute group II afferents. At the onset of our studies, the primary goal was to uncover definitive molecular markers for MS and GTO afferent proprioceptors, and we were excited to find more molecular clusters than the 2-3 we had anticipated. Based on the notion that clusters 1 and 5 represent group Ia MS afferents and group Ib GTO afferents, respectively, we concluded that the remaining clusters (2-4) correspond to group II afferents. We concede that our data does not fully support this conclusion given that we, due to technical limitations, were unable to directly link group 2-4 marker expression to a group II afferent identity. In the current version of the manuscript we acknowledge this caveat and explain our conclusions more carefully.

As per the reviewer's suggestion we have nevertheless tried very hard to gain additional insights into the correlation between these molecular subtypes and muscle afferent identity and acquired additional antibodies (Pdlm [clusters 1-3], Asb2 [clusters 3,4], Kcnh8 [clusters 2-4], Kv1.6 [cluster 4]) that potentially would allow us to locate these markers to a particular type of sensory ending. Unfortunately, despite these efforts, these experiments did not yield any conclusive results.

We also considered expanding our segmental specific RNA *in situ* analysis for cluster 2-4 markers but decided against this for two reasons. With the exception of several cluster 4 markers (e.g. *Cdh13*, *Tac1*), cluster 2 and 3 markers are not exclusively expressed in their bioinformatics assigned cluster (see Figure 2g). For instance, *Irak4* transcript level is highest in clusters 2 and 3, but this marker is also observed in clusters 1, 3, and 5; *Thzh1* and *Fam196a* are expressed in clusters 2 and 4, and *Asb2* in clusters 3 and 4. In addition, while markers specific for limb muscle afferents may be selective for specific segmental levels, markers for body wall or axial muscle afferents are expected to be found in all ganglia along the rostro-caudal extent of the spinal cord. Based on these observations, we are unlikely to find all-or-none distributions for any of the putative type II clusters, and as such, we think it would be impossible to draw any further conclusions from these experiments beyond what we have currently reported.

Related to this, the authors propose that the two functional classes of proprioceptors detected in their *in vitro* physiology experiments correspond to group I versus group II afferents, but this is not tested. *In vivo* physiological measures would be most impactful in addressing this possibility as well as addressing the issue about the presumed type II proprioceptor diversity, mentioned above. In light of the genetic tools that this study describes, one could try to specifically label Group Ia afferents using Calb2-Cre; Runx3-FlpO and determine if proprioceptors labeled this way display the rapidly adapting responses observed in some of their cells when all proprioceptors all labeled. Similarly, *Tac1*-Cre could be used intersectionally with Runx3-FlpO to label group II afferents and determine whether or not they show a tonic firing pattern, as predicted.

This issue has been raised by all three reviewers. We agree that in the absence of electrophysiological recordings of molecularly defined proprioceptors there is a need for caution in drawing conclusions.

To address this issue, we now provide additional electrophysiological recording experiments that are in support of our initial conclusion. In our original analyses, we identified two major pSN firing patterns

following sustained current injection: phasic (single AP and burst responses) and tonic. We speculated that, based on prior observations in the literature, the neurons with phasic responses corresponded to groups Ia and Ib afferents, while tonic neurons represented group II afferents.

We also described that the voltage-gated Kv1.1 and Kv1.2 potassium channels are expressed at elevated levels in group I [cluster 1 and 5] afferents when compared to presumed group II [cluster 2-4] afferents. We now have been able to demonstrate that application of alpha-Dendrotoxin (DTX α ; an inhibitor of Kv1.1/Kv1.2 channels) consistently reverts the phasic response to a tonic firing pattern in response to current injection. In contrast, tonic responses are largely unaffected by DTX α . These findings not only suggest that the differential expression of these potassium channels contributes to the unique features of proprioceptor firing properties, but also support the idea that group Ia and Ib afferents comprise afferents that exhibit dynamic firing properties. These data are now added to the reconfigured Figure 6 and Supplementary Figure 11.

In response to the reviewer's suggestions, we also attempted to perform patch-clamp recordings on tdT labeled group Ia afferents (indeed using an intersectional labeling strategy based on the *Calb2:Cre;Rx3:FlpO;Ai65:tdT* reporter). Despite multiple attempts - dissociating these neurons from several animals using a range of dissociation methods - we were unable to perform high quality physiological recordings due to the sparsity and unhealthy nature of the tdT⁺ neurons. Two of the cells that were nevertheless patched revealed a burst-firing phenotype, but given the unhealthy status of the cells we feel it is premature to include these data.

We hypothesize that the difficulty to acquire high quality recordings from whole-cell patch experiments from these neurons stems from two (related) reasons. First, while we find that over 85% of spindles are innervated by a Calretinin⁺ afferent (based on immunohistology), genetically-labeled tdT⁺ spindles (using the *Calb2:Cre;Rx3:FlpO;Ai65* mice) are only observed in a subset of muscles. For instance, in EDL and ECR muscles most or all spindles are marked by a tdT⁺ afferent, but in Soleus or Plantaris muscles, few if any, spindles comprise a tdT⁺ Ia afferent. This discrepancy between immunological and genetic labeling is consistent for a given muscle. At present we do not fully understand the basis of the inefficient genetic labeling. It may relate to the absence/misplacement of certain enhancer elements in the *Calb2:Cre* line (due to the Cre insertion into the locus), or may reflect a muscle-by-muscle difference in CR expression levels in Ia afferents, such that the Ai65 reporter is inefficiently activated in muscles in which CR (and hence Cre) is expressed at lower levels. Regardless of the reason, these observations effectively reveal that MS Ia afferents consist in different flavors: those that are labelled by *Calb2:Cre* and those that are not. We further speculate (reason 2) that the afferents that *are* labeled by tdT (and which we have tried to record from) are particularly sensitive for mechanical manipulation thus resulting in reduced survival during dissociation procedures.

Lastly, we want to clarify that use of the *Tac1:Cre* allele does not permit the selective labeling of cluster 4 neurons. As we demonstrated (Supplementary Figure 8b), it appears that at early developmental stages *Tac1* is transiently expressed in most sensory neurons, including proprioceptors (and not just cluster 4 afferents). Therefore, even when using the *Tac1:Cre* allele in an intersectional approach with *Runx3:FlpO* (expressed in all proprioceptors), we will not be able to distinguish between proprioceptors that are labeled as a consequence of the transient developmental expression and those labelled because of expression of *Tac1:Cre* in the adult. *Tac:CreErt2* animals would be a solution to this problem, but we are not aware of the existence of this mouse line currently.

Specific comments:

1. Although the work presented is generally of high quality, some of the findings, including the extent of overlap between identified marker genes by in situ is described in qualitative terms (“generally co-expressed,” “often overlaps with”) and supported by representative images (Fig. 3, Supplemental Figure S5). Quantification is limited to the % total tdT⁺ neurons positive for each marker gene. Adding the quantification of the % overlap between putative subset-specific markers would greatly improve this analysis.

The RNA scope data in Figure 3 serves to provide evidence that the key molecules identified through single cell RNA sequencing are in fact expressed in subpopulations of proprioceptors, and as such, our quantification focused on the relationship between each marker and expression of tdT.

We agree with the notion that additional quantification of the overlap between individual markers would provide added value. Indeed, we initially performed the counts as the reviewer suggested: besides determining the prevalence of each marker within the tdT⁺ population we also examined the overlap between the two cluster markers analyzed. In most cases (due to constraints inherent to the RNAscope protocol) the combination of markers involved Pcdh8 (as it is one of the few more reliable markers we have that can be used in the third fluorescent channel). As indicated in Figure 2g, in addition to cluster 5 neurons, Pcdh8 is also expressed at lower levels in many cluster 2-4 neurons. Thus, these analyses meant that we were examining co-expression between cluster 1 and cluster 2-5 neurons rather than between clusters 1 and 5. We felt that these data would be of incremental use. We also considered - and performed - some of these counts in relation to expression of PV (as a surrogate marker for proprioceptors). However, we feel that these analyses are also not really informative. This is because PV is expressed in many neurons outside the tdT⁺ proprioceptor population (~25%), and we cannot be certain if any co-expression between specific cluster markers is in proprioceptors or in low threshold cutaneous mechanoreceptors. Despite these considerations, we have added all prior generated co-expression data to Supplementary Figure 5.

2. In a similar fashion, more quantification to demonstrate the efficiency and specificity of newly identified genetic tools would be helpful in determining their usefulness for various types of experiments. For example, the authors show that Calb2-Cre; LSL-GFP labels many but not all muscle spindles: what % of spindles are labeled? Similarly, Tac1-Cre is shown to label afferents at the polar ends of muscle spindles (type II) but it is noted that it can also label afferents at the equatorial regions of spindles (type Ia): at what frequency are these two classes observed? What proportion of all type II afferents does Tac1-Cre efficiently label?

We agree. We now provide quantitative data for CR⁺ and vGlut2⁺ MS afferents across a wide range of muscles in Supplementary Tables 1 and 2, respectively. We also provide quantitative data on genetically labeled CR⁺ afferents using *Calb2:Cre;Rx3:FlpO;Ai65* mice (included in Supplementary Table 1). As described above, these data indicate that the vast majority of all MSs (85.7%; 276/322 MSs) contain at least one CR⁺ group Ia afferent when assessed through immunohistological analyses, while the utility of the *Calb2:Cre* driver is limited to labeling group Ia afferents in a subset of muscles. GTO's are nearly never innervated by CR⁺ or afferents (1%; 1/102 GTOs).

We have now also added counts for the relative numbers of presumed groups Ia and II afferent tdT⁺ fibers in a few muscles in an *Tac1:Cre;Ai14:tdT* animal (Supplementary Table 3). We reiterate here that

these data are of relative limited value considering the early transient expression of *Tac1:Cre* in many proprioceptors beyond cluster 4 neurons. We do note, however, that, proportionally, tdT is more often observed in group II afferents than in group Ia afferents (taking into account the overall larger number of group II afferents), suggesting that expression at later stages is biased to group II afferents.

Finally, we want to note that numbers of groups Ia, II or Ib afferents remain extremely difficult to quantify accurately given that the ‘assignment’ of an afferent as a group I vs group II based on morphological criteria requires whole mount muscle staining. Whole mount muscle stains for adult muscle are technically challenging – due to difficulties with antibody/light penetration – especially in larger muscles. Based on these limitations, our data sample of muscles is biased to more accessible muscles (e.g. EDL, Soleus, Plantaris, ECR, Gluteus). An additional complication for body wall, axial, and intercostal muscle is that muscles are not organized in discrete units. For these muscles we examined all spindles in the muscle segment used.

3. It may be beyond the scope of the current study to characterize the contributions of different ion channels to the firing properties of proprioceptors observed *in vitro*, but one way to test the author’s hypothesis that the different potassium channels expressed in type I vs type II proprioceptors contribute to different intrinsic firing properties would be to use pharmacology to test the contribution of some of the candidate channels identified (for instance Kv1.1 and Kv1.2, which are more highly expressed in clusters 1 and 5, putative type I proprioceptors, and may contribute to their rapid adaptation to current injection). This would improve this section of the study, which otherwise remains descriptive and speculative.

As described above (major point #2), we have now added these experiments (see new Figure 6).

4. The sequence analysis for the adult population relies on 208 neurons, which is not a large number for this type of work. The developmental analysis relies on even fewer neurons than the adult analysis (in particular at P0, 55 neurons) and it is therefore difficult to be confident that the determination of fewer clusters truly reflects decreased molecular diversity at this age and is not simply due to lower resolution in cluster identification.

We agree that 208 neurons is not a large number but we emphasize that this is a nearly pure population of proprioceptors with very high quality sequencing data (when compared to other studies profiling sensory neurons or proprioceptors). We also note that our original population of neurons was 450 but we applied very stringent quality controls (i.e. we eliminated nearly half of our neurons due to satellite cell contamination), which ultimately reduced the final number of neurons for analysis to 208. In addition, our choice of platform (plate seq) enables the detection of far more genes than is typically achieved through platforms optimized for large cell numbers (i.e. 10x genomics or drop-seq) (see Supplementary Fig. 4a, 4b). We note that this relatively high gene detection rate is also observed for all our developmental datasets when compared to prior studies using a 10x platform (see Figure 1 below; we would be happy to include this data in the manuscript if deemed useful). We also would like to reiterate that a similar observation - that sensory neurons are transcriptionally more diverse at later developmental time points when compared to earlier stages – was described by Sharma et al. Nevertheless, with respect to the molecular diversity at the embryonic stages we acknowledge that we may need to be more careful with our conclusions as it is possible that upon sequencing higher numbers

of cells, additional molecular diversity may be uncovered, in particular with respect to proprioceptor muscle-type identity.

Figure 1. Comparison of single cell sequencing platforms. Mean number of genes and transcription factors/neuron in p0 proprioceptors/sensory neurons as detected using a plate-based sequencing platform (Oliver et al. – this manuscript) or a droplet-based sequencing platform (Sharma et al., 2019).

5. The authors assume that early *Pou4f3* (*Brn3c*) expression predicts future type Ib identity, but do not test this. A lineage tracing experiment would strengthen the resulting arguments and interpretation of the developmental sequencing data.

In *PV:Cre;Pou4f3-AP* mice, expression of AP is under direct control of the *Pou4f3* promoter and is activated by *PV:Cre* mediated excision of a transcriptional STOP sequence. *PV* expression (and as such *Cre*) in proprioceptors is first detected around e13-14, thus activating AP expression in (all) *Pou4f3*⁺ proprioceptors from this age onward. (Note that after *Cre*-mediated excision of the STOP sequence, AP expression in *PV* neurons is controlled by *Pou4f3* promoter elements.) As we demonstrated, robust AP activity can be detected in GTO afferent endings at p2 (see Figure 4i), while AP activity/*Pou4f3* expression in MS afferents barely remains detectable at this point (see Supplementary Fig. 7b).

To demonstrate that *Pou4f3* is in fact expressed in both MS and GTO afferents at earlier developmental stages, we have now also provided data from e17.5 *PV:Cre;Pou4f3-AP* muscle, where the AP labeling is evident in both MS and GTO sensory endings (see Supplementary Fig. 7c). In further support of this conclusion, we note that in spinal cord, ventral horn-projecting afferents (typically assumed to represent group Ia MS afferents) are abundantly present at p0 but nearly extinct at p12 (Supplementary Fig. 7d).

6. Is Supplementary Fig. 2g missing?

We apologize for the confusion. Supplemental Figure 2g was omitted from the submitted version of our manuscript. We have now removed all references to Supplemental Figure 2g.

7. The abstract states: “definitive molecular correlates of these three muscle afferent subtypes remain unknown” and the Introduction states “definitive molecular signature for functionally distinct proprioceptive muscle afferents remains to be established” While this is true, defining molecular diversity as an endpoint may not be of sufficient impact for a research article, especially without definitive proof of their relationship to neuronal populations with defined physiological, functional or morphological properties, although it may be for a resource paper if it provides clear genetic tools for these populations and new molecular insights.

In this manuscript we describe the molecular diversity of proprioceptive muscle afferents and provide definitive morphological evidence that cluster 1 afferents correspond to group Ia MS afferents and that cluster 5 afferents represent group Ib afferents (Figure 4). We also identify and characterize several new proprioceptor selective genetic resources for the scientific community. We extended these observations by offering new insights into the electrophysiological features of proprioceptors and examine how these features correlate with expression biases of specific ion channels (Figure 5). Finally, our data demonstrates that mature proprioceptor subtypes diverge after they innervate their receptor targets through a protracted developmental process (echoing previous findings by Sharma et al., (2019) published in Nature). We acknowledge that we may be biased in our assessment, but we believe these data represent a significant step forward in the field.

Reviewer #2 (Remarks to the Author):

This manuscript reports the first evidence of unique molecular signatures for different classes of muscle proprioceptors. These specialized mechanoreceptors are indispensable to kinesthetic sense as well as to correction of and prediction about sensory-guided movement, yet the specific roles of different classes of muscle proprioceptors in these processes remain incompletely understood, as do cellular mechanisms involved in their sensory encoding and network operations. Findings reported here introduce a new experimental tool/strategy having real potential to substantively improve understanding in these fields. The key result is demonstration that unique clusters of molecules predict the identities of proprioceptor sub-classes independently validated by morphological features of the sensory receptors they innervate. These determinations were made with rigorous and creative applications of genetic strategies coupled with single-cell RNA sequencing and bioinformatics approaches

that uncovered molecular/histological means for labeling and distinguishing proprioceptor subtypes. In a particularly significant observation arising out of the first application of this approach, lineage analysis at different developmental stages leads the authors’ to suggest that the unique transcriptional profiles of the proprioceptor subtypes are induced by contact with their receptor targets. Experimental design and rigor are carefully detailed, the primary and supplemental data and analyses are sound, incisive and convincing, the findings are clearly presented, properly supported by relevant literature, and carefully interpreted in general. A few comments are offered below suggesting reasons for tempering some

conclusions and providing more detailed presentation of some data. These are relatively moderate flaws that do not diminish the likelihood that delineation of muscle proprioceptors reported here will prove extremely valuable to studies aimed at determining their roles in control of sensory-guided movement and more specifically, their contributions to neural networks, and specificity in regeneration.

Major comments

- My major comment has to do with statements and conclusions claiming the data show that neurons with different molecular expression correspond to or were tested for being functionally distinct subtypes (see Abstract, Intro summary page 5, Subtitle page 9, text pages 11, 12). These statements over-reach the data. Functional responses to mechanical stimulation were not measured but only inferred from the anatomy of receptor innervation and terminal endings. The 3 molecular clusters assigned to group II may or may not be functionally distinct, and all muscle proprioceptors – Ia, II, spindle unclassified and Ib – exhibit some functional redundancy. Comments and conclusions related to assigning molecular signature to afferent function require more precision and nuance.

We acknowledge our mistake in conflating morphologically distinct subtypes with functional subtypes. It was never our intention to give the impression we examined our molecularly defined subclasses at a functional level. We have now amended the manuscript to restrict any correlations between molecular identities to morphologically defined subtypes.

- The whole-cell electrophysiological recordings of membrane and firing properties have value in describing the range physiological properties of PVRx3;tdT neurons. However, there are no data demonstrating how slowly vs rapidly adapting neurons were associated with neuronal subclass or with different potassium channels, which might have been examined from voltage-clamp study K currents. For these reasons, speculation in Results (top paragraph p. 14) should be limited and restricted to Discussion.

We have addressed this issue above (reviewer #1, major point 2) but repeat it here for convenience:

we now provide additional electrophysiological recording experiments that are in support of our initial conclusion. In our original analyses, we identified two major pSN firing patterns following sustained current injection: phasic (single AP and burst responses) and tonic. We speculated that, based on prior observations in the literature, the neurons with phasic responses corresponded to groups Ia and Ib afferents, while tonic neurons represented group II afferents.

We also described that the voltage-gated Kv1.1 and Kv1.2 potassium channels are expressed at elevated levels in group I [cluster 1 and 5] afferents when compared to presumed group II [cluster 2-4] afferents. We now have been able to demonstrate that application of alpha-Dendrotoxin (DTX α ; an inhibitor of Kv1.1/Kv1.2 channels), consistently reverts the phasic response to a tonic firing pattern in response to current injection. In contrast, tonic responses are largely unaffected by DTX α . These findings not only suggest that the differential expression of these potassium channels contribute to the unique features of proprioceptor firing properties, but also support the idea that group Ia and Ib afferents comprise afferents that exhibit dynamic firing properties. These data are now added to the reconfigured Figure 6 and Supplementary Figure 11.

- Data demonstrating that a neuron's expression of unique molecules predicts the receptor or receptor region it innervates is the central finding. It is important then to provide data on the number of neurons and their receptor connections examined per cluster and the number of neurons that connected with the 'wrong' receptor. These interests may even warrant a table.

We agree. We have now added these data for Calretinin (CR) and vGlut2 protein expression in Supplementary Tables 1 and 2 (and Supplementary Figure 6). As explained above, counts on the number of *Tac1:tdT⁺* afferents are provided (Supplementary Table 3) but of limited use considering the imperfect genetic reagents. In addition, while we believe *Pou4f3* marks all GTO afferents (based on the typical pattern of GTOs in many of the muscles we analyzed), it is not feasible (due to incompatibility with the alkaline phosphatase staining reaction) to independently verify whether this truly reflects all GTOs using another independent generic GTO-afferent marker (e.g. vGlut1). Staining for *Pdh8* is unfortunately not strong enough to reliably assess its expression in a range of different adult muscle targets.

- It is arguable that the authors assign too much significance to the diversity in MS subtypes. Diversity in molecular identification fits well with the continuum observed in the spike train encoding properties of Ia, II, and unclassified MS afferents. The appearance of 3 group II clusters may reflect the large range in mechanosensory encoding by group II, assuming that the 3 group II clusters are actually distinguishable physiologically. The observation have more to do with imprecision/arbitrariness in physiological classification than is does with meaningful differences.

We agree and we have now deemphasized this in the text when appropriate.

Minor comments

Page 6: "... A few afferents that appear to project to deep dorsal horn targets, ...". This statement needs revision recognizing that group Ia, II and Ib all send dense synaptic projections into deep dorsal horn (LV, VI) in cat (Brown and Fyffe, 1978) rat (Vincent et al. 2017), and mouse (Levine et al. 2014).

We regret this inaccuracy as we are aware that spinal targets of proprioceptors include dorsal horn neurons. We have now corrected our descriptions of these afferents – emphasizing their lateral entry into the spinal cord as opposed to the medial trajectory typically followed by proprioceptive afferents.

Reviewer #3 (Remarks to the Author):

The manuscript by Katherine Oliver and colleagues presents the first thorough, complete and detailed single-cell transcriptomic analysis of proprioceptive afferents in the adult mouse and also during embryonic and early postnatal development. Proprioceptor afferents are critical for movement and the sense of self and space, and although they have been thoroughly studied physiologically, their molecular/genetic identify has remained largely unknown. This is in part because, as described in this study, their unique properties arise not due to specific genetic lineages and early fating, but are regulated by extrinsic factors during postnatal development. Thus, they do not have a defined specific genetic code that could be used for their identification throughout development. This has been suspected for some time now in the field, but this study finally makes the full and complete case for this fact. The work now defines their genetic differences in the adult after developing their specific properties and constitutes a tour-de-force with impressive validation of many of the genes

identified. It is full of new findings that are the culmination of this very long effort by several labs seeking to uncover the molecular identity of these afferents. Where others failed the lab of Dr. Joriene de Nooij succeeded, and in a very impressive manner.

The main new findings to highlight are that: 1) proprioceptors that are usually classified in three physiological types (Ia, Ib, II) can in fact be divided in 5 molecular/genetic classes considering three subclasses of type II afferents adding diversity to these important and seldom studied afferent type; 2) revealing for the first time clear molecular markers to distinguish Ia and Ib afferents, with some being quite surprising (VGLUT2 in Ia afferents); 3) finding that this genetic/molecular diversity arises during maturation of these afferents, i.e., different proprioceptive afferents are not derived from specific lineages but arise by developmental factors influencing their maturation after they arrive to muscle (this confirms a previous report from this same group suggesting that key proprioceptive genes are dependent on muscle factors); 4) the study uncovers a diversity of K channel expression patterns that may underlie the physiological differences among these afferents; 5) finally, an intriguing afferent-type specific diversity in GABAA receptor subunit expression was also found suggesting that mechanisms that control the transfer of sensory information from each of these afferents to the CNS could be controlled by presynaptic mechanisms that differ in pharmacology and kinetics and could be dissected out in future studies. These results are all extremely significant and will be highly used and likely cited by future investigators. They also provide entry points for developing new animal models to specifically dissect the individual functional significance of each of these afferents. This has not yet been possible. If the results and conclusions from this study are confirmed and further validated by others, it is my view that this study will constitute one of the major advances in the study of proprioception in the last decade. In any case this should be a valuable resource for future research and the whole data set will be made available as indicated in the manuscript (although the exact details are not yet defined) (Page 12: An interactive web application to search for individual genes in proprioceptor subtypes will be made available at xxxx).

However, there are also significant issues that need to be fixed. Many are detailed in a point by point manner below. However, *the major weakness of this study is the poor correlation between the physiological data and the molecular subtypes identified.* It is unclear why a direct study revealing the molecular identity of the recorded cells was not done. The more indirect and speculative correlations proposed might be valid, but they make a much less compelling case for the conclusions proposed. Moreover, since the two major firing types described are not directly correlated with any of the differences in channel expression revealed by the genetic data.

We believe we have addressed this issue in response to similar comments by reviewers #1 and #2, and reiterate that we now have added data that appears to correlate Kv1.1/Kv1.2 potassium channel activity with electrophysiological features of group I subtypes.

In addition, there are several other significant issues throughout the manuscript that require the authors attention. These are detailed in the following points.

1. Counting method: Different figures show data for either number of DRGs or number of sections. It is unclear why there should be two different ways of sampling data. Why not analyze data always per

DRG? Since sectioning increases sampling variability, this approach would diminish the large variability observed in many graphs. This variability sometimes goes from 0% to 100%, which it can be interpreted as the sections being highly variable in the number of cells they contain for analysis. Setting the data points per section seems not representative of the real distribution. Can all figures be set per DRG analyzed? Methods indicate that minimum of 3 DRG were sampled per experiments (with a few exceptions). These should be enough for statistical comparisons when enough sections per DRG are studied to obtain accurate DRG data points.

DRG counts are inherently complicated as the distribution of the neurons is not equal across the DRG. In particular for markers present in low numbers of neurons, counts can change significantly across sections. We agree with the reviewer that total DRG counts (counting at least 50% of all DRG sections evenly sampled from across the entire DRG) are preferable as they provide a more accurate measure of the number of positive cells, and this is why we have preferred this method whenever feasible (i.e. for neuronal counts of genetically labeled neurons which we can readily identify with reliable reagents).

For RNA scope we used a section-based counting approach because we considered it more relevant to know the average distribution of each marker across the tdT population (rather than per DRG), as well as for logistical reasons. Total DRG RNAscope counts are prohibitively time consuming (at least on the scale of this manuscript; in many cases the signals are too weak for low power imaging thus requiring stitching of a composite adult L4/5 DRG section from 3-4 individual images).

2. Page 6 (also figure S1f): Projections to dorsal horn laminae do not rule out that an axon could be from a proprioceptive Ia afferent. Distal flexors project to dorsal horn laminae in the trajectory shown in the figure (see Figure 9 in: Ishizuka N, Mannen H, Hongo T, Sasaki S. 1979 J Comp Neurol.186(2):189-211. doi:10.1002/cne.901860206).

As discussed above (reviewer #2, minor comments), this mistake is now corrected.

3. Page 6: "Intriguingly, we only observed tdT+ Merkel cell afferents in forelimb skin and not in the skin of the back."

What about hind limbs which are the focus of the paper?

We focused our initial analysis on forelimb skin since forelimb paws are innervated by a relatively higher density of cutaneous mechanoreceptors, thus offering an increased chance we would detect any putative tdT afferents outside of muscle afferent population. We have now also performed these analyses and added the new data for hindlimb paws. We indeed detected tdT+ neurons that associate with Merkel cells in touch pads and digit skin (see updated Figure 1e).

4. Page 7: "PVRx3:tdT reporter provides an efficient and nearly exclusive means to manipulate or isolate proprioceptive muscle afferents."

Can this be supported with quantitative data? What percentage of PVRx3:tdT DRG cells are not proprioceptors?

This issue is difficult to address since we do not have molecular markers for the non-proprioceptive PVRx3+ population that are distinct from PVRx3+ proprioceptive muscle afferents. In the absence of such markers we can only obtain this information from determining how many neurons terminate outside

muscle – a nearly impossible task. Upon inspection of Rx3 and PV expression in tdT⁺ neurons, however, we find that $2.7 \pm 1.1\%$ of tdT neurons (in L4/5 DRG; $0.8 \pm 0.8\%$ in L1-3 DRG; data now included in Supplementary Figure 2) express no or near undetectable levels of PV transcript. These PV⁺tdT⁺ neurons, which perhaps only transiently expressed PV, possibly could represent the non-muscle afferent tdT neurons. (We note that all 5 bioinformatics identified clusters of neurons analyzed in the manuscript express high levels of PV.) In addition, mature proprioceptors are known to express the PDZ-domain protein Whirlin, but we noticed that for one of the small clusters not included in our analysis (7/208 cells or 3.4%) none of the cells expressed Whirlin, suggesting that this population may also account for some of the non-muscle afferent tdT⁺ neurons. It remains unclear if the Whirlin⁺tdT⁺ neurons and PV⁺tdT⁺ neurons represent similar or divergent neuronal subsets. Importantly, in absence of definitive markers for these neuronal subsets these analyses, unfortunately, remain speculative.

5. Page 8: "Expression of Cdh13 and Crtac1 is biased to neonate proprioceptors that innervate dorsal (flexor) or ventral (extensor) limb muscle targets."

The correspondence between flexor and extensor actions and ventral vs dorsal limb muscle is full of exceptions and contradictions and should be abandoned. The gastrocnemius is ventral but is a flexor of the knee and an extensor of the ankle; the biceps is a flexor at the knee but ventral, the rectus femoris is a knee extensor also a hip flexor and dorsal. The vastus medialis is a pure knee extensor but dorsal ... and so on. The developmental significant issue for axon trajectories and wiring is the location of developing the muscles, not whether they function as flexors/extensors across different joints. I understand this generalized equivalences became standard in the field, but that they keep being maintained is very perplexing.

We are in total agreement on this issue. We have now amended this point and describe embryonic muscles only in terms of their anatomical location.

6. Page 8: this patterns (not plural).

Typo – now corrected.

7. Page 10: "The pattern of CR⁺ Ia afferents in adult muscle raises the possibility that not all muscle spindles are innervated by a group Ia afferent."

This conclusion is quite a "stretch" since there is no other evidence provided for this possibility. Moreover, it is repeated in conclusions. The authors should be careful about this and provide additional supportive data since if this was true it would be a major finding. This should be easily tested by adding an alternative marker of Ia afferents in the spindle (VGLUT1 annulospiral endings around the central bag fibers). The experiment should be done before making such a dramatic conclusion. The alternative possibility of diversity in expression of CR by Ia afferents seems better supported by the single cell expression data in Figure 2g.

It is generally agreed upon that MSs consist of one Ia afferent and up to several type II afferents that flank the Ia terminal. In reality, spindles can vary dramatically, ranging from highly complex spindles with multiple afferents (sometimes including multiple primary endings), to spindles where there is only one sensory ending. We find that 86% of spindles we examined (276/322, across ten different muscles) are innervated by a CR⁺ afferent, which exhibits a highly stereotypic spiral morphology that is far more

regular than most other afferent fibers that innervate the spindle. This afferent is typically located at the equatorial region (judging from the diameter of the intrafusal muscle fibers or from the flanking CR⁺ afferent terminals). Based on these observations we concluded that these morphologically unique and CR⁺ expressing afferents correspond to group Ia afferents. This indicates that approximately 15% of spindles lack a CR⁺ afferent. For some of these spindles the absence of a CR⁺ fiber may reflect that the spindle is too deep in the tissue to allow efficient antibody (or light) penetration for analysis/detection. However, a subset of these spindles simply do not possess a sensory terminal that resembles the CR⁺ afferent; in other words, not only do they lack a CR⁺ afferent, they also lack an afferent with the typical CR⁺ afferent morphology. We interpreted this to mean that some spindles may not be innervated by a group Ia afferent (as defined based on generally agreed upon morphological definitions). Nevertheless, we concede that a more definitive analysis of this issue requires counterstaining with a marker for the bag1 fiber and we now leave any such speculation to the discussion.

8. Page 11: "Consistent with previous observations that most DRG neurons express Pou4f3 at early embryonic stages, initially all pSNs (i.e. both MS and GTO afferents) are labelled by AP (Supplementary Fig. 7a-c)."

I do not think I follow the data presented in here. The data shown in Fig S7 is all at p2 and it seems that muscle spindles express little or any Pou4f3. Is there data at any other earlier time point that supports this statement of all pSNs expressing Pou4f3 before p2?

As discussed above, we now also performed these experiments at an earlier stage (e17.5) and demonstrate that, along with group Ib GTO afferents, Ia afferents initially show abundant Alkaline Phosphatase activity. We have now included these data in Supplementary Figure 7.

9. Page 14. "Based on previous observations that groups Ia and Ib afferents have dynamic (RA) response properties, we postulate that RA neurons correspond to groups Ia and Ib proprioceptors, while tonic neurons correspond to group II afferents."

I find this not very compelling; the dynamic and static responses to stretch and force of Ia, Ib and II afferents are encoded by ion channels at the terminal end in the spindle or GTO and not by ion channels at the cell body. As adequately discussed later in conclusions, Ia and Ib can evoke significant amounts of tonic firing to sustained stretch or tension. In muscle spindles the tonic firing of Ia afferents to sustained stretch can be as high or even higher than that of type II. The main differences are the dynamic responses. For a description of the the firing properties of these afferents in rodents consult the characterization of Vincent JA, Gabriel HM, Deardorff AS, et al. 2017J Neurophysiol.;118(5):2687-2701. doi:10.1152/jn.00497.2017. Considering this it seems necessary to obtain direct evidence that the neurons with single spikes have molecular features of Ia or Ib afferents and the one with tonic firing through the pulse represent type II.

That will be a very interesting finding and as discussed later parallels findings in Merkel afferents. However comparisons with Merkel afferents to justify Ia/Ib identity breakdowns in logic because Merkel afferents responses to skin indentation are slowly adapting and therefore comparison with Ia/Ib I(considered by the authors RA) vs II (considered by the authors as SA) is not straightforward. These problems could be solved by experimentation that directly reveals the identity of the recorded cells.

We'd like to respond to this comment in sequence:

1. “the dynamic and static responses to stretch and force of Ia, Ib and II afferents are encoded by ion channels at the terminal end in the spindle or GTO and not by ion channels at the cell body.”

While this is true *in vivo*, several prior studies have indicated that cell bodies of proprioceptors and Merkel cell afferents can respond to mechanical stimulation *in vitro*, demonstrating that at least certain aspects of the mechanoreceptive machinery and associated regulatory components can be relocated to the cell body when the axons are severed (e.g. Woo et al., 2015; Zheng et al., 2019).

2. “However comparisons with Merkel afferents to justify Ia/Ib identity breakdowns in logic because Merkel afferents responses to skin indentation are slowly adapting and therefore comparison with Ia/Ib I(considered by the authors RA) vs II (considered by the authors as SA) is not straightforward.”

It has been demonstrated that Merkel cell afferents, in absence of functional Merkel cells (lacking Piezo 2), exhibit phasic response properties (see Woo et al, 2014; Maksimovic et al., 2014). We proposed the parallels to Merkel cell afferents on the basis of these reports.

3. “These problems could be solved by experimentation that directly reveals the identity of the recorded cells.”

As described previously, we now are adding data that indicates that the firing properties of the phasic (RA) neuronal subset can be influenced by the Kv1.1/Kv1.2 inhibitor DTX α : changing a phasic firing pattern into a tonic firing pattern (new Figure 6). Given that clusters I and 5 neurons (representing groups Ia and Ib afferents, respectively) expressed the highest levels of these potassium channels, these data offer additional support for the idea that these afferents correspond to neuronal subsets characterized by a phasic firing pattern.

10. On VGLUT2 expression in Ia afferents: “However, the observation that adult group Ia MS afferents MS, in addition to vGlut1, also express vGlut2 was unexpected and may have important implications for studies that assess inputs to spinal neurons on the basis of the presumed differential expression of these two glutamate transporters between proprioceptors and spinal neurons.”

I find this statement problematic in the sense that without qualification will create tremendous confusion. Quantification of VGLUT1 and VGLUT2 expression in the synaptic boutons of electrophysiologically identified and intracellular filled Ia afferents reported that both were present at significant levels above background, but while VGLUT1 was detected at >10 times background levels, VGLUT2 was barely above background (Alvarez FJ, Titus-Mitchell HE, Bullinger KL, Kraszpulski M, Nardelli P, Cope TC. J Neurophysiol. 2011;106(5):2450-2470. doi:10.1152/jn.01095.2010). Moreover, high sensitive EM ICC using postembedding colloidal gold in cryosubstituted specimens reported that many VGLUT1 synapses in the spinal cord also contain low levels of VGLUT2, but this was more difficult to detect using co-localization with immunofluorescence (Alvarez FJ, Villalba RM, Zerda R, Schneider SP. J Comp Neurol. 2004;472(3):257-280. doi:10.1002/cne.20012). Thus, VGLUT2-IR boutons detected with immunofluorescence can hardly be traced to Ia afferents, contrary to what is suggested here by the authors. VGLUT2 levels in the terminal might be too low. However, it might explain why neurotransmission is still preserved in the monosynaptic stretch reflex in VGLUT1 KOs. Mende M, Fletcher EV, Belluardo JL, et al. 2016Neuron.;90(6):1189-1202. doi:10.1016/j.neuron.2016.05.008.

We regret not being more careful with our statements on this issue. Based on our observation that group Ia afferents at p12 and p56 appear to express significant levels of vGlut2 transcript and protein (Figures 2g and 4g-iii), we assumed that the protein would also be present in the central terminals of these afferents (similar to vGlut1). Reconciling our observation with the evidence/prior reports cited by the reviewer, we consider two possibilities. First, it is possible that vGlut2 protein is only minimally transported centrally. Second, vGlut2 expression levels may vary among group Ia afferents (and generally remain lower than vGlut1 levels) and it is possible that the sample of afferents assessed previously (rat MG, LG, Soleus) did not include muscles innervated by MS afferents with more prominent vGlut2 protein expression (e.g. EDL). We have now amended the text and no longer refer to the implications for proprioceptor assignment based on spinal vGlut expression profiles.

11. Related to the above discussion, it is quite common for VGLUT1 systems to switch during development from VGLUT2 to VGLUT1. This has not yet been fully shown for proprioceptors. Does the data suggest that this is the case? Moreover was VGLUT1 detected? Is it possible that VGLUT2 low level expression in Ia afferents is a result of incomplete developmental downregulation?

Our developmental data indicates that at earlier stages (e14, P0) vGlut2 is indeed more prevalent (e14) and more abundant (p0) than vGlut1. At p12, as predicted by the reviewer, we see a reversal: vGlut1 is expressed in essentially all proprioceptors while vGlut2 is mostly restricted to presumptive groups Ia and Ib afferents. At even later stages (p56), vGlut2 expression is largely eliminated from all but group Ia afferent proprioceptors, while vGlut1 remains abundantly expressed in all proprioceptors. Since we also observe abundant expression of vGlut2 in the Ia peripheral endings in adult muscle, we consider it unlikely that this pattern of vGlut2 expression is merely a consequence of an incomplete developmental downregulation. We have included a screenshot of our interactive web based widget (see Figure 2 below; accessible to all readers at the time of publication of the manuscript) to illustrate these observations.

Figure 2. Developmental expression of *vGlut1* (*Slc17a7*) and *vGlut2* (*Slc17a6*) in bioinformatics-identified proprioceptor subtypes.

Figures/Figure legends issues:

12. Figure 1. I am not clear about the necessity of including S100 immunolabelings. I think it is distracting from the main point.

S100 immunolabeling was intended to help identify the sensory end organs of the skin-innervating tdTomato positive afferents. We agree with the reviewer that in these examples the S100 labeling does not provide additional value. We have now removed the S100 channel from the images.

13. Figure S1e. "while fragments of GFP+PV+ neurons (lacking a nucleus) can be detected (arrowheads), 'full size' (nucleated) GFP+PV+ neurons that lack RX3 are not observed." I am not clear what the authors are referring to in here and arrowheads are not shown.

We apologize for any confusion. In our copy of Supplemental Figure 1e (overview image on the left) there were (two yellow) arrowheads but they may not have been as clearly visible as needed. To increase visibility, we have now replaced the arrowheads with (larger) arrows. Both arrows are directed at small PV⁺GFP⁺ segments that lack a visible Runx3⁺ nucleus. These segments are clearly smaller in size than the average PV⁺GFP⁺ cell bodies that invariably do contain a Rx3⁺ nucleus. We consider these data an indication that few - if any - PV⁺Rx3^{off} neurons (e.g. Meissner afferents) are marked by GFP due to an early developmental phase of Rx3:FlpO expression. (These data are also supported by our analysis of the peripheral endings in Supplemental Figure 3i, demonstrating that Meissner, Pacinian, or Lanceolate endings do not label with tdTomato.)

14. Figure S1f,g I would remove the box graphs indicating averages and percentiles since these stats make no sense with $n = 2$. It says that $n = 2$ DRG, how many sections on average to estimate each DRG data point? Methods say that all sections were analyzed. Please clarify in figure legend.

We agree. We now provide bar plots. DRG data points were not estimated but based on total DRG counts, i.e. entire DRG were sectioned (at 30 μm) and every section was counted. We have now provided the exact number of sections/DRG in the figure legend.

15. Figure S1g. I do not see the need to show the percentage of PV+ and PV- Rx3+ neurons. It is a binary criterion that sums 100% therefore the data in both box plots is identical, just the inverse of each other. You only need to show one of them.

We have removed the PV- data from the plot.

16. Figure S1i. Please refer to this figure in the appropriate section in the text. RPV- and RPV+ are confusing. Legend refers them as Rx3+PV+ and Rx3+PVoff. It will be best to be consistent in labeling between figure and figure legend.

We apologize for any confusion. We now refer to this figure in the text. We also have revised the Figure to increase clarity and to ensure consistency in the labeling

17. Figure S5. The image panels are frequently mismatched with the descriptions in the results. Please review.

We apologize. In the revised manuscript we have corrected these mistakes.

18. Figure 4j. Is the GTO labeled by VGLUT1(-ICC) as in figure or GFP+ GTO as in legend? If the second, how is that accomplished: VGLUT1-GFP?

The GTO is labeled by vGlut1 as indicated in the Figure – the legend has now been corrected.

19. Figure S4. In c, 173 cells are indicated as distributed in the five clusters. But adding all the cells in f, the number is 166. Why the discrepancy?

The correct number is 166. The number 173 was a mistake left over from an earlier draft of the manuscript. We thank the reviewer for identifying this error.

Also throughout the manuscript it is never explained the depiction of the different clusters in figures like the f panel. What does it mean the overlapping regions between C1 and C2, C3 and C4 and C4 with C5?

There is no overlap between any of these clusters, the apparent overlap stemmed from our desire to have equal space for the captions/legends for each clusters (i.e. cluster C2 is rather small and it would be difficult to fit the “15 (9%)” addition above the column). We have now revised each of these bar plots so that the cells are more clearly delineated with respect to the cluster they belong to.

20. Figure 2 vs S4. The % of cells in C1 is 31% in Figure S4 and 30% in Figure 2b. Are there small

differences in the number of analyzed cells between these figures?

There are no differences between these figures. The correct percentage for cluster C1 is 30.7%. In Figure S4 we rounded this to 31%. Since we also rounded up the percentages for most other clusters (C2 = 9.0%, C3 = 14.7%, C4 = 16.9%, C5 = 18.7%) the total number in Figure S4f is 101%. The calculation for C1 in Figure 2b was based on subtracting the percentages of all other groups from the total 100% - this resulted in 30%. We have now corrected the data in Figure 2e to 31% given this is the more accurate number.

21. Figure 3i (ii) the legend indicates that expression of *Nxph1* and *Pcdh8* in this cell is comparable, but that is clearly not the case in the figure. That is not the case in Figure S4g, either. The conclusion that comparable expression of these two genes defines C2, C3, C4 seems not to be accurate in view of the data with RNAscope and the expression profile in Figure 2g.

We agree. We meant to say that C2-C4 cells express both *Nxph1* and *Pcdh8*, but that levels of *Pcdh8* are variable (as expressly illustrated in Supplementary Figure 4g). We have now corrected the wording in the figure legend.

22. Figure S6f,g. Both spindles display genetically labeled VR axons flanked by VGLUT1-IR endings in more polar regions. Was one of them supposed to show an example of a GFP+ axon not flanked by non-GFP+ VGLUT1-IR axons as in legend.

We apologize that the order of the panels was inconsistent with the legend. We have now amended these figure panels. The point of these panels is to demonstrate that many CR⁺ afferents occupy the equatorial region of the spindle (by virtue of being flanked by two non-CR⁺vGlut1⁺ afferents (as in *gi*)), but that there is considerable morphological variability across spindles and spindle afferents (i.e. some CR⁺ afferents are not flanked by other vGlut1⁺ afferents; as in *gii*).

23. Figure S6i, What is the evidence that the genetically labeled CR axon belongs to a gamma motoneuron? Are there any end plates to show this axon is not sensory? If is indeed motor, could it be a beta axon?

Our evidence for this is that the CR:GFP⁺ terminal (in Figure S6i) that innervates the spindle does not co-express vGlut1. We consider vGlut1 a reliable marker for all spindle innervating afferents, which (as far as we are aware) is not expressed in (α , β , or γ) motor neurons. We concede that the motor neuron may also be a beta axon (we did not trace it back to determine if it also forms endplates on extrafusal fibers).

24. Figure 6c. Calbindin in motor axons. Calbindin is expressed by embryonic neurons and quickly downregulated during postnatal development but the time course of downregulation is pool dependent. In this sense the image in Figure 6c from a P12 might be more and exception than the norm. Please indicate so in the figure legend to avoid misleading the reader. Calbindin-ICC very infrequently labels NMJs. Moreover, the relevance of these data in this paper is not clear.

We agree that any data not directly contributing to the main message of the paper may be unnecessary distracting. We have now removed these panels to allow additional space for quantitative data on CR and vGlut2 afferents.

References:

Maksimovic S, Nakatani M, Baba Y, Nelson AM, Marshall KL, Wellnitz SA, Firozi P, Woo SH, Ranade S, Patapoutian A, Lumpkin EA. Epidermal Merkel cells are mechanosensory cells that tune mammalian touch receptors. *Nature*. 2014 May 29;509(7502):617-21. doi: 10.1038/nature13250. Epub 2014 Apr 6. PMID: 24717432; PMCID: PMC4097312.

Sharma N, Flaherty K, Lezgiyeva K, Wagner DE, Klein AM, Ginty DD. The emergence of transcriptional identity in somatosensory neurons. *Nature*. 2020 Jan;577(7790):392-398. doi: 10.1038/s41586-019-1900-1. Epub 2020 Jan 8. PMID: 31915380; PMCID: PMC7307422.

Woo SH, Ranade S, Weyer AD, Dubin AE, Baba Y, Qiu Z, Petrus M, Miyamoto T, Reddy K, Lumpkin EA, Stucky CL, Patapoutian A. Piezo2 is required for Merkel-cell mechanotransduction. *Nature*. 2014 May 29;509(7502):622-6. doi: 10.1038/nature13251. Epub 2014 Apr 6. PMID: 24717433; PMCID: PMC4039622.

Woo SH, Lukacs V, de Nooij JC, Zaytseva D, Criddle CR, Francisco A, Jessell TM, Wilkinson KA, Patapoutian A. Piezo2 is the principal mechanotransduction channel for proprioception. *Nat Neurosci*. 2015 Dec;18(12):1756-62. doi: 10.1038/nn.4162. Epub 2015 Nov 9. PMID: 26551544; PMCID: PMC4661126.

Zheng Y, Liu P, Bai L, Trimmer JS, Bean BP, Ginty DD. Deep Sequencing of Somatosensory Neurons Reveals Molecular Determinants of Intrinsic Physiological Properties. *Neuron*. 2019 Aug 21;103(4):598-616.e7. doi: 10.1016/j.neuron.2019.05.039. Epub 2019 Jun 24. PMID: 31248728; PMCID: PMC6706313.

Reviewer #2 (Remarks to the Author):

This manuscript provides information, which, as the authors suggest, "is long overdue" concerning distinctive molecular signatures provisionally corresponding to the three classic categories of muscle proprioceptive neurons. The tool/approach for distinguishing proprioceptors has the potential to significantly expand information about the distribution of afferent types to different spinal networks. Molecular distinction will also prove useful in studies of adaptive/plastic responses and development, for which the authors provide new evidence suggesting a dependence of transcriptional expression upon contact with receptor targets. The manuscript retains its original strengths in rigorous experimental design, execution, and analysis, which generate high quality data and careful interpretation that yield convincing conclusions. The authors' responded to critiques with new data and edits that adequately address reviewer concerns and strengthen the paper. For example, adding pharmacological studies helps associate differences in ion-channel expression and firing behavior to cluster identification.

Reviewer #3 (Remarks to the Author):

The authors have resolved the majority of the issues from the previous review. The major criticism raised by all three reviewers was the lack of direct physiological confirmation of the type of afferent corresponding to each of the five subgroups of molecularly identified proprioceptors. Although two types, C1 and C5, clearly correspond morphologically to primary spindle afferent terminals (Ia) and Golgi tendon organs (Ib), the nature of C2, C3 and C4 was assumed as type II, based on exclusionary information since their terminations in muscle could not be directly visualized. The authors now have taken a more careful approach interpreting these as "possible" but not definitive type II afferents. Nevertheless, at least for some C4 afferents they provide evidence that they end as secondary spindle afferent terminals, suggesting a type II nature. Physiologically they recorded in vitro their firing responses to depolarizing pulses and concluded that phasic phenotypes correspond to type I and tonic phenotypes to type II afferents. They have now expanded their electrophysiological experiments to confirm that phasic firing phenotypes depend on expression of Kv1.1 and Kv1.2, two channels enriched in the two molecular groups (C1 and C5) confirmed morphologically as Ia and Ib afferents. This ameliorates previous criticisms; however, it is noted that "group II" C2, C3, C4 afferents also seem to have some expression of these channels according to Figure 5a. As it is now pointed out the exact nature of the three putative type II groups must wait for direct recordings of their sensory encoding capabilities, in particular given the transitional phenotypes between Ia and II afferents reported in the literature and pointed out by a previous reviewer. Despite the fact that this issue is not yet fully settled in the manuscript the present version contains a significant amount of very significant and convincing information on the molecular nature of Ia and Ib fibers and the possible diversity of the type II group all derived from very high-quality analyses. They also provide important information in the relatively late time-point at which these different phenotypes develop, contrasting to other neuronal types fated at very early stages of their development, frequently soon after they emerge from progenitors. All these constitute a tremendous advance in the field. This work is also quite timely and, in my opinion, publication should not be delayed further at this point. It seems to me that setting up for physiologically recordings of molecular identified afferents will now take a very long time preventing the scientific community of being informed of the many important advances already contained within the manuscript. The authors have modified their conclusions now more in accordance with the data presented and they clearly point out to existing gaps.

Two very minor editorial issues need to be corrected.

- Supplemental Figure 4 C. Please correct the 173 number from the figure legend.
- Main text line 210. Following on the authors "reviewer reply" they should delete "flexor/extensor identity" in this line since this is not really investigated with the markers used.

Response to reviewer's comments:

We again want to thank both reviewers for their positive comments on our revised manuscript entitled "Molecular correlates of muscle spindle and Golgi tendon organ sensory afferents revealed by single proprioceptor transcriptome analysis". We also want to reiterate our appreciation for the continued efforts of both reviewers for their help in improving the manuscript and by alerting us to the remaining mistakes. Our responses to the outstanding two issues are below.

(Reviewer #3)

- Supplemental Figure 4 C. Please correct the 173 number from the figure legend.
This issue has now been corrected.
- Main text line 210. Following on the authors "reviewer reply" they should delete "flexor/extensor identity" in this line since this is not really investigated with the markers used.
We agree. We have now (really) omitted this from the text.